# RAS/PI3K pathway mutations sensitise epithelial ovarian cancer cells to a PARP/NAMPT inhibitor combination
Michael Gruet [1,2,5], Yitao Xu[1,5], Lyutong An [1], Yurui Ma [1], Cristina Balcells[1], Katie Tyson[1], Laila C. Evangelista [1], Sarah Spear[1], Yuewei Xu[1,3], Flora McKinney[1], Julia Babuta[4], Chandler Bray[1], Chiharu Wickremesinghe[1], Alexandros P. Siskos [1], Anke M. Nijhuis[1], Edward W. Tate [2], Iain A. McNeish [1], Adrian Benito[1] ✉ & Hector C. Keun [1] ✉

The combination of PARP and NAMPT inhibitors (PARPi/NAMPTi) has been explored for the treatment of triple-negative breast cancer, Ewing sarcoma and high-grade serous carcinoma (HGSC). However, dose limiting toxicity has hampered NAMPTi in clinical trials. To maximise the therapeutic window, we set out to identify predictive genomic biomarkers. Bioinformatic analysis and screening of a panel of epithelial ovarian cancer (EOC) cell lines revealed that cells with RAS/PI3K pathway mutations are sensitive to the NAMPTi FK866. Combined exposure to olaparib and FK866 is associated with a reduction in nicotinamide mononucleotide (NMN) and the PARP substrate nicotinamide adenine dinucleotide (NAD$^+$), with coincident increases in ROS production, DNA damage and apoptosis induction. Caspase 3/7 activity is upregulated to a greater extent in RAS/PI3K mutant cell lines. Finally, the combination significantly reduces omental tumour weight and increases overall survival in mice injected with ID8 *Trp53$^{-/-}$;Pten$^{-/-}$* cells. This study highlights the potential of the PARPi/NAMPTi combination in RAS/PI3K pathway mutant EOC.

Ovarian cancer is the second most commonly diagnosed gynaecological cancer in the UK[1], and is the most common cause of gynaecological cancer deaths[2]. The most common ovarian cancer subtype, high-grade serous carcinoma (HGSC), frequently harbours mutations in *BRCA1/2*, leading to a homologous recombination deficiency (HRD). HRD tumours are exquisitely sensitive to inhibitors of poly(ADP-ribose) polymerase (PARP), such as olaparib[3,4]. PARP inhibitors (PARPi) are approved in BRCA-mutated (BRCAm) patients for front-line maintenance therapy and maintenance in the recurrent setting[4]. However, evidence is emerging suggesting that patients could benefit from PARPi-therapy independent of BRCAm status[5]. Despite this, the majority of patients will relapse within 3 years while on PARPi maintenance[6], highlighting the need to identify combination therapies to tackle treatment resistance.

PARPs regulate cellular processes, such as DNA repair, by catalysing post-translational ADP-ribosylation. This involves the covalent attachment of one or more ADP-ribose units onto targets proteins, using nicotinamide adenine dinucleotide (NAD$^+$) as a substrate[7]. The majority

of NAD$^+$ is supplied through the NAD$^+$ salvage pathway, and the rate limiting step is catalysed by the enzyme nicotinamide phosphoribosyl-transferase (NAMPT). NAMPT converts nicotinamide into nicotinamide mononucleotide (NMN), which is then used to synthesise NAD$^{+8}$. Because of the link between NAD$^+$ metabolism and PARP catalytic activity, groups have investigated whether the use of a NAMPT inhibitor (NAMPTi) and PARPi combination could be beneficial. Encouragingly, PARPi/NAMPTi combinations have shown synergistic activity in triple-negative breast cancer (TNBC)[9], Ewing's Sarcoma[10], and HGSC[11]. However, NAMPTis, such as FK866[12], have not proved successful as monotherapies in clinical trials largely because of toxicity[13]. Considering this, there is value in identifying predictive biomarkers to improve the therapeutic window and reduce NAMPTi-associated toxicity[13]. To date, only *BRCA1/2* loss has been identified as a potential biomarker for the combination[9].

Interestingly, a post-hoc exploratory biomarker analysis from the ARIEL2 study revealed that rucaparib-treated patients harbouring

---

[1]Department of Surgery and Cancer, Imperial College London, London, UK. [2]Department of Chemistry, Molecular Sciences Research Hub, Imperial College London, London, UK. [3]Wisdom Lake Academy of Pharmacy, Xi'an Jiaotong-Liverpool University, Suzhou, China. [4]Department of Metabolism, Digestion and Reproduction, Burlington Danes, Imperial College London, London, UK. [5]These authors contributed equally: Michael Gruet, Yitao Xu. ✉e-mail: a.benito-mauricio@imperial.ac.uk; h.keun@imperial.ac.uk

alterations in genes involved in the RAS pathway or PI3K/AKT signalling ('RAS/PI3K-mutant') had poorer outcomes when compared to other groups[14]. RAS proteins exert their functions through several effector pathways, particularly PI3K signalling[15]. Given these pathways are frequently mutated in up to 45% of HGSC cases[16], and in other EOC subtypes[17], there is a need to identify therapeutic strategies that can extend the benefit of PARPis within this group of patients.

Hyperactivation of RAS/PI3K signalling results in uncontrolled proliferation and increased demands on cellular metabolism. In this context, $NAD^+$ recycling must increase to sustain high proliferation rates, because $NAD^+$ plays an essential role as a coenzyme in key cellular metabolic pathways, such as glycolysis and the TCA cycle[18]. Interestingly, a relationship between the $NAD^+$ salvage pathway and RAS/PI3K signalling is emerging. *KRAS*-mutant tumours have been shown to possess lower intracellular $NAD^+$ concentrations[19,20], reflecting increased metabolic demand. *PTEN* expression inversely correlates with *NAMPT* expression[21], and oncogenic $BRAF^{V600E}$ mutations can induce *NAMPT* overexpression, leading to NAMPTi sensitivity[22,23]. Considering this, we hypothesised that RAS/PI3K-mutant EOC cells have an increased demand for $NAD^+$ that can be exploited with a NAMPTi, and consequently may exhibit greater responses to treatment with a PARPi/NAMPTi combination. To address this, we assessed the efficacy of a PARPi/NAMPTi combination in vitro using a panel of EOC cell lines, and in vivo using the ID8-$Trp53^{-/-}$; $Pten^{-/-}$model[24,25].

Our experiments demonstrate that RAS/PI3K-mutant EOC cells are sensitive to FK866 and benefit from combined treatment with olaparib and FK866. The combination has anti-cancer activity in a murine model of HGSC with a *Pten* deletion. Together our data demonstrate the utility of a PARPi/NAMPTi combination in EOC models with genomic alterations in RAS/PI3K pathways and suggest these mutations could be used as a predictive biomarker.

## Results

### RAS/PI3K pathway mutant EOC cell lines are sensitive to the NAMPT inhibitor FK866

To investigate the role of NAMPT in EOC, data from the Genomics of Drug Sensitivity in Cancer (GDSC) resource (https://www.cancerrxgene.org/) were analysed. This analysis demonstrated that ovarian cancer cell lines are not intrinsically sensitive to the NAMPTi FK866 (Fig. 1a). However, when ovarian cancer cells were segregated based on their *KRAS* or *PI3K* (includes *PIK3CA* and *PTEN* mutations) mutational status (Fig. 1b), FK866-sensitivity ($IC_{50}$) was more comparable to the most sensitive cell types (highly sensitive blood cancers (Fig. 1a)). A significant difference in FK866-sensitivity could be observed between wildtype and *KRAS/PI3K*-mutant ovarian cancer cells ($P = 0.0355$ and $P = 0.0379$, respectively).

To validate these findings, a panel of ten ovarian cancer cell lines was assembled, with genomic features that reflect different EOC subtypes[26] (Fig. 2a). The panel included six RAS/PI3K-wildtype cell lines. COV318, COV504 and 59M cells have *TP53* mutations. KURAMOCHI and OVSAHO cells possess *TP53* and *BRCA2* mutations, and it should be noted that KURAMOCHI cells are reported to have amplified *KRAS*[26]. COV644 cells do not possess mutations in these pathways. Four RAS/PI3K-mutant cell lines were included in the panel. A2780 cells possess *BRAF*, *PI3K* and *PTEN* mutations. HEY-A8 cells have *BRAF* and *KRAS* mutations. OVCAR-8 cells possess *TP53*, *ERBB2* and *KRAS* mutations, but have been reported to have a heterozygous *BRCA1* methylation[27]. Finally, TOV21G cells have *KRAS*, *PI3K* and *PTEN* mutations. To assess their relative FK866 sensitivity, the panel was treated with the NAMPTi for 6 days and $IC_{50}$ values were calculated (Fig. 2b and Supplementary Fig. 1a). This confirmed that the four RAS/PI3K-mutant cell lines were more sensitive to NAMPT-inhibition than the RAS/PI3K-wildtype cell lines. Western blotting was performed in several RAS/PI3K-wildtype (COV318 and COV644) and -mutant (A2780 and TOV21G) cell lines to assess whether MAPK/ERK and PI3K/AKT

**Fig. 1 | A subset of ovarian tumours are sensitive to the NAMPTi FK866. a** Using the Genomics of Drug Sensitivity in Cancer (GDSC) resource (https://www.cancerrxgene.org/) FK866 (NAMPTi) sensitivity was characterised in different cancer cell lines. Log $IC_{50}$ values are shown for each cell line in a box and whisker plot, individual points are shown in blue. Ovarian cancer is highlighted in yellow. **b** The log $IC_{50}$ values observed in KRAS-mutant (Mut), PI3K-mutant (*PIK3CA* and *PTEN* mutations) and -wildtype (wt) EOC cell lines are shown. Statistical significance was determined using the brown-forsythe and welch ANOVA tests, with Dunnett's t3 multiple comparisons test.

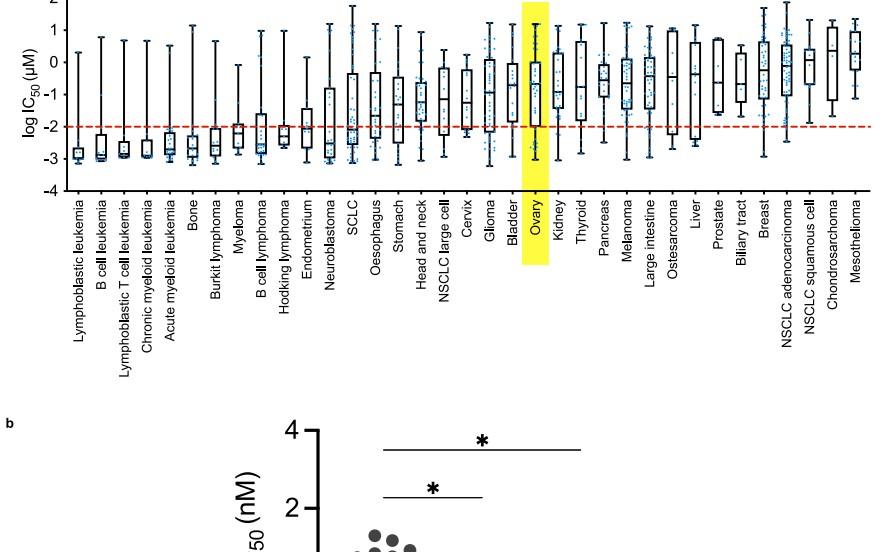

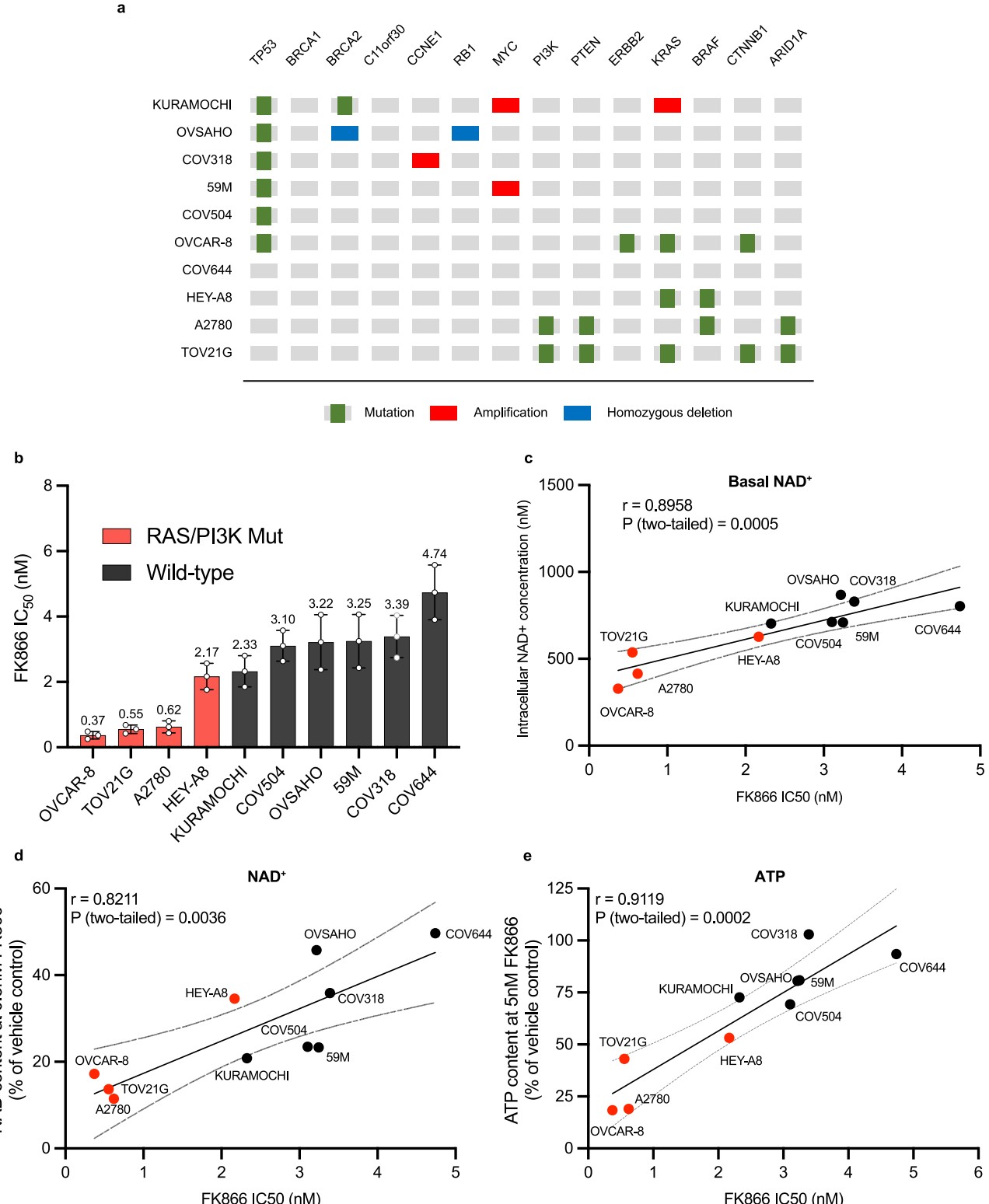

**Fig. 2 | RAS/PI3K pathway mutant EOC cell lines are sensitive to the NAMPT inhibitor FK866. a** Mutation profiles of selected epithelial ovarian cancer cell lines, this graphic has been adapted from Domcke et al. (2013)[26]. **b** EOC cell lines FK866 sensitivity after 6-days treatment. Cell biomass was measured using the SRB assay. $IC_{50}$ values are shown for each cell line, along with their 95% confidence intervals. **c** Basal $NAD^+$ concentration (nM). **d** $NAD^+$ content (% of VC) after 24-h treatment with 0.5 nM of FK866. **e** ATP content (% of VC) after 48-h treatment with 5 nM of FK866. **c–e** Data were normalised to cell biomass (SRB assay). Pearson correlation coefficient (r) and linear regression were used to analyse the relationship between $NAD^+$/ATP content with FK866 and FK866 sensitivity ($IC_{50}$). r and p values are shown.

signalling pathways are active (Supplementary Fig. 1b). AKT phosphorylation was detected in the A2780 and TOV21G cell lines. ERK1/2 phosphorylation was also assessed, and, excluding TOV21G cells, was detectable in all cell lines.

Basal NAD$^+$ concentrations were assessed in each cell line (Fig. 2c), interestingly, the smallest NAD$^+$ pools were observed in the RAS/PI3K-mutant cell lines (A2780, HEY-A8, OVCAR-8 and TOV21G). Furthermore, the NAD$^+$ pool size correlates with the FK866 sensitivity (IC$_{50}$) (r = 0.8958, p = 0.0005). To investigate whether increased FK866 sensitivity in RAS/PI3K-mutant cell lines was due to increased depletion of NAD$^+$ pools, cells were treated with FK866 for 24-h and NAD$^+$ assays were performed (Fig. 2d and Supplementary Fig. 1c). When treated with a low dose of FK866 (0.5 nM), clear differences emerged. The three most sensitive RAS/PI3K mutant cell lines (A2780, OVCAR-8 and TOV21G) achieved the greatest reduction in their NAD$^+$ pools (>82% reduction), whilst a ~65% reduction was achieved in the less-sensitive HEY-A8 cell line. Interestingly, the level of NAD$^+$ depletion was found to correlate with FK866 sensitivity (IC$_{50}$) (r = 0.8211, p = 0.0036) (Fig. 2d). To gain further insight into the impact of FK866 on NMN and NAD$^+$ pools, a UPLC-MS/MS metabolomics assay was performed using COV318 (RAS/PI3K-wildtype) and OVCAR-8 (RAS/PI3K-mutant) cells (Supplementary Fig. 2a, b). This revealed that COV318 cells had larger NMN (~336% larger) and NAD$^+$ (~298% larger) pools when compared to OVCAR-8 cells under basal conditions. Furthermore, 24-h treatment with 1 nM FK866 reduced NMN and NAD$^+$ pools to a greater extent in OVCAR-8 cells (91% and 96%, respectively) when compared to COV318 (~51% and 68%, respectively).

NAMPT inhibition with FK866 inhibits ATP synthesis, which leads to the delayed induction of cell death, as NADH is a substrate for the mitochondrial electron transport chain and NAD$^+$ is a cofactor required for glycolysis[28,29]. To investigate the impact of FK866 treatment on ATP levels, the EOC panel was treated for 48-h with FK866, and an ATP assay was performed (Fig. 2e and Supplementary Fig. 1d). This revealed 5 nM FK866 could reduce ATP levels in the majority of EOC cell lines tested. However, the greatest reductions were observed in the RAS/PI3K-mutant cell lines (>47%). Interestingly, the level of ATP depletion also correlated with FK866 sensitivity (IC$_{50}$) (r = 0.9119, p = 0.0002) (Fig. 2e).

### Combined PARP and NAMPT inhibition is more effective in RAS/PI3K pathway mutant 3D spheroids

We evaluated whether FK866 could enhance PARPi responses in the EOC panel in 2D culture (Fig. 3a and Supplementary Fig. 3). In this experiment, cells were treated for 6-days with olaparib and low doses of FK866 (100pM or 500pM) that did not impact cell proliferation as a monotherapy. Due to their increased FK866-sensitivity A2780, OVCAR-8 and TOV21G cells were only co-treated with 100pM FK866. In all cell lines tested co-treatment with FK866 reduced olaparib IC$_{50}$ values. Bliss synergy maps (6 ×6) were also produced in select RAS/PI3K-wildtype (COV318, KURAMOCHI, OVSAHO) and RAS/PI3K-mutant (A2780, HEY-A8 and OVCAR-8) cell lines (Supplementary Fig. 4). The strongest levels of synergy were observed in the RAS/PI3K-mutant A2780 (Supplementary Fig. 4d) and OVCAR-8 (Supplementary Fig. 4f) cell lines.

Since 3D models better recapitulate the complexity of the in vivo tumour microenvironment[30], COV318 (Fig. 3b and Supplementary Fig. 5a), A2780 (Fig. 3c and Supplementary Fig. 5b), HEY-A8 (Fig. 3d and Supplementary Fig. 5c) and OVCAR-8 cells (Fig. 3e and Supplementary Fig. 5d) growing in 3D were treated for 6-days with olaparib and/or FK866, before assessing cell viability. COV644 and TOV21G cells were not included as they do not form spheroids. Interestingly, co-treatment with FK866 significantly improved PARPi responses in all three RAS/PI3K-mutant cell lines, but not in the RAS/PI3K-wildtype COV318 cell line.

Since inhibiting PARP activity is known to decrease NAD$^+$ consumption[31], we also evaluated the influence of the PARPi/NAMPTi combination on NMN and NAD$^+$ pools using a UPLC-MS/MS metabolomics assay (Supplementary Fig. 2a, b). This revealed that 24-h treatment

with 1 μM olaparib leads to a slight increase in NMN and NAD$^+$ levels in COV318 but not in OVCAR-8 cells. However, when cells were co-treated with olaparib and FK866, the NAMPTi-induced reduction of NMN/NAD$^+$ pools far outweighed any reduction in NMN/NAD$^+$ consumption caused by the inhibition of PARP activity (ADP-ribosylation).

### Combined PARP and NAMPT inhibition induces apoptosis by depleting NMN and NAD$^+$ in RAS/PI3K pathway mutant EOC cell lines

Next, we assessed the influence of olaparib, FK866 and the combination on apoptosis induction in COV318 (Fig. 4a and Supplementary Fig. 6b), COV644 (Fig. 4b) A2780 (Fig. 4c and Supplementary Fig. 6c) and TOV21G cells (Fig. 4d and Supplementary Fig. 6d). Whether assessing apoptosis/necrosis using Annexin V staining (24-h) or a caspase 3/7 activation assay (48-h), FK866 and/or olaparib had little-to-no influence on apoptosis in the RAS/PI3K-wildtype COV318 and COV644 cell lines. However, in RAS/PI3K-mutant A2780 and TOV21G cells, a significant increase in apoptosis/necrosis (Supplementary Fig. 6c, d) and caspase activity (Fig. 4c, d) could be observed following treatment with the combination. To confirm that FK866 potentiates PARP-inhibitor responses by depleting the NMN/NAD$^+$ pool, a rescue experiment was performed, where medium was supplemented with NMN during drug treatment and caspase 3/7 activity was evaluated (Fig. 4e, f). As expected, supplementing medium with NMN rescued A2780 cells treated with the combination (Fig. 4f), and caspase 3/7 activity was comparable to cells treated with olaparib alone.

### ROS-generation promotes the induction of apoptosis following combined PARP and NAMPT inhibition in RAS/PI3K pathway mutant EOC cell lines

Since increased reactive oxygen species (ROS) generation and loss of mitochondrial membrane potential (MMP) is often observed in FK866-induced cell death[28,29], we evaluated the influence of combined PARP and NAMPT inhibition on ROS levels and MMP. Olaparib and FK866 monotherapy had very little influence on ROS levels in RAS/PI3K-wildtype (COV318 and COV644) (Fig. 5a, b) and RAS/PI3K-mutant (A2780 and TOV21G) (Fig. 5c, d) cell lines. However, when A2780 and TOV21G cells were treated with the combination, a significant increase in ROS could be observed (Fig. 5c, d), with the largest increase observed in A2780 cells (Fig. 5c). Next, we assessed the influence of monotherapy and combination treatment on loss of MMP in COV318 and A2780 cells (Fig. 5e, f). Both monotherapy and combination treatment had no influence on MMP in COV318 cells (Fig. 5e). However, in A2780 cells, loss of MMP was observed in all conditions (Fig. 5f). The largest loss of MMP was observed in A2780 cells treated with the combination (4.5-fold increase), and this was followed by cells treated with FK866 (3.8-fold increase) and olaparib alone (2.7-fold increase). To evaluate the influence of ROS generation on apoptosis/necrosis-induction (Fig. 5g and Supplementary Fig. 6a) and caspase 3/7 activity (Fig. 5h), rescue experiments were performed in A2780 cells, where medium was supplemented with the ROS-scavenger N-acetylcysteine (NAC)[32]. Interestingly, NAC supplementation rescues apoptosis/necrosis (Fig. 5g), but the increased caspase 3/7 activity observed following combination treatment was only partially rescued (Fig. 5h).

### Combined PARP and NAMPT inhibition upregulates DNA damage in RAS/PI3K pathway mutant EOC cell lines

Since the combination has previously been shown to induce apoptosis by upregulating DNA damage[9–11], we next evaluated the influence of the olaparib/FK866 combination on γH2AX foci formation (Fig. 6a). In COV318 cells (RAS/PI3K-wildtype) olaparib and/or FK866 treatment had very little influence on γH2AX foci (Fig. 6b). Whereas, in TOV21G cells (RAS/PI3K-mutant) combination treatment led to a significant increase in the number of γH2AX foci (Fig. 6c).

Given NAMPTi[23,33] and PARPi[34] can induce G2/M arrest as single agents, we set out to evaluate the influence of the combination on the cell

**Fig. 3 | FK866 potentiates the cytotoxic effects of olaparib in EOC cell lines but is more effective in RAS/PI3K pathway mutant 3D spheroids. a** EOC cell lines were co-treated in 2D with olaparib and FK866 (0.1 nM or 0.5 nM) to assess their sensitivity to the combination after 6-days treatment. Cell biomass was measured using the SRB assay. $IC_{50}$ values are shown for each cell line, along with their 95% confidence intervals. **b–e** 3D spheroids were co-treated with olaparib (1 μM or 10 μM) and FK866 (0.5 nM or 1 nM) to assess their sensitivity to the combination after 6-days treatment. Cell viability was measured using the CellTitre-Glo® 3D assay. Data are the average ± SD of three independent experiments. Statistical significance was determined using 2-way ANOVA followed by Tukey's multiple comparisons test ($p < 0.05$, **$p < 0.01$, *** $p < 0.001$, **** $p < 0.0001$).

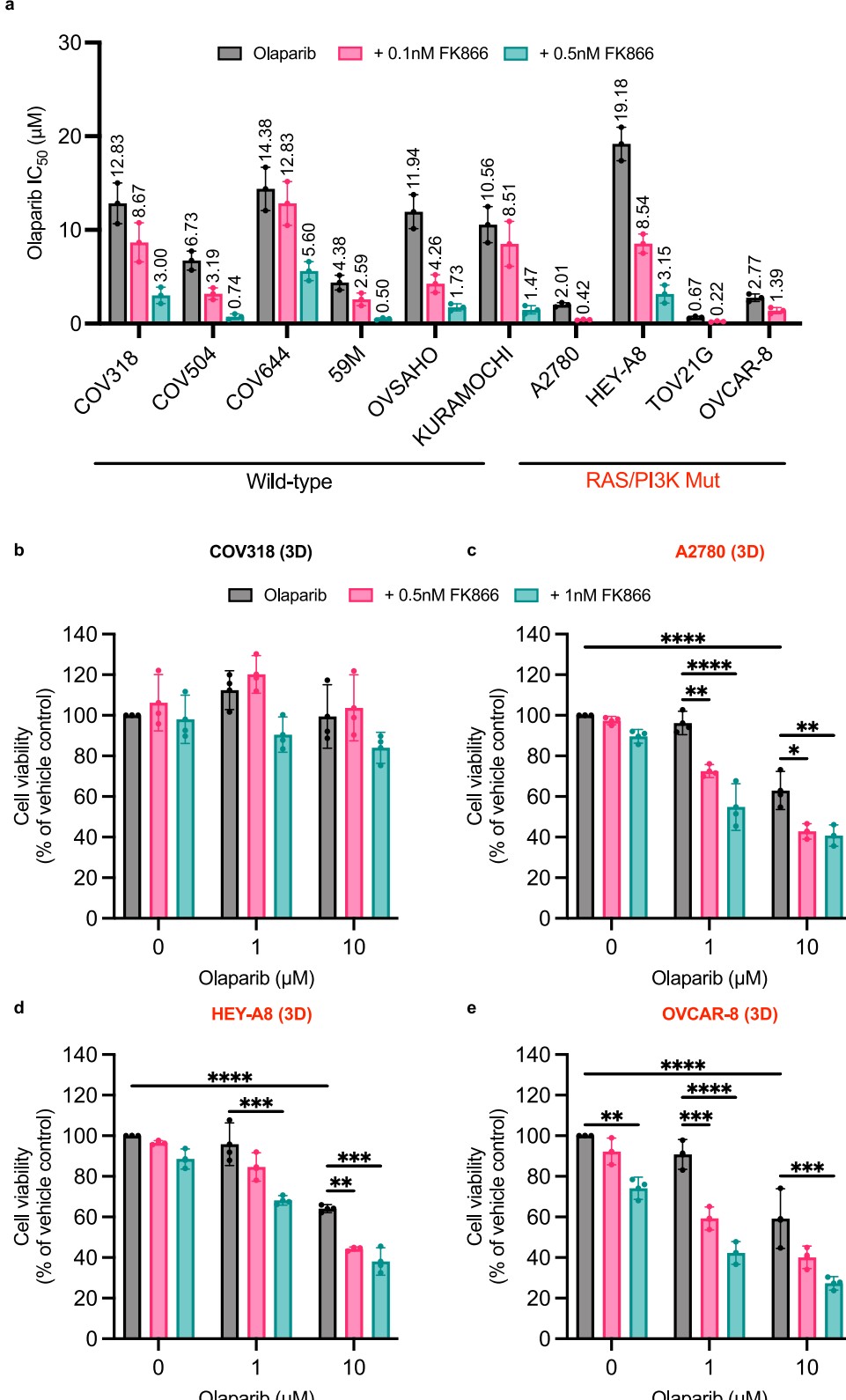

cycle (Supplementary Fig. 7). In COV318 cells (Supplementary Fig. 7b), olaparib and/or FK866 treatment had no influence on the proportion of cells in the G2/M phase. Whereas, in A2780 (Supplementary Fig. 7c) and TOV21G (Supplementary Fig. 7D) cells, both monotherapy and combination treatment increased the proportion of cells in the G2/M phase, the latter induced the largest increase.

**Combined PARP and NAMPT inhibition is effective in vitro and in vivo in the ID8-*Trp53*^−/−; *Pten*^−/− model**

Since HGSC is the most common subtype of EOC[17] and alterations in RAS/PI3K-signalling[14], such as *PTEN* loss/mutations[35], may worsen PARPi-responses in the clinic, we investigated whether the combination of a PARP and NAMPT inhibitor would have activity in a transplantable

**Fig. 4 | Combination treatment upregulates caspase 3/7 activity in RAS/PI3K pathway mutant EOC cell lines. a–d** COV318, COV644, A2780 and TOV21G cells were co-treated for 48-h with olaparib (10 μM) and FK866 (5 nM) to assess their influence on caspase 3/7 activity (apoptosis). This was measured using the Caspase-Glo® 3/7 assay, data was normalised to cell biomass (SRB assay). **e, f** A caspase-rescue experiment was performed in COV318 and A2780 cells by supplementing medium with 250 μM NMN. Data are the average ± SD of three independent experiments. Statistical significance was determined using two-way ANOVA followed by Tukey's multiple comparisons test ($p < 0.05$, **$p < 0.01$, *** $p < 0.001$, **** $p < 0.0001$).

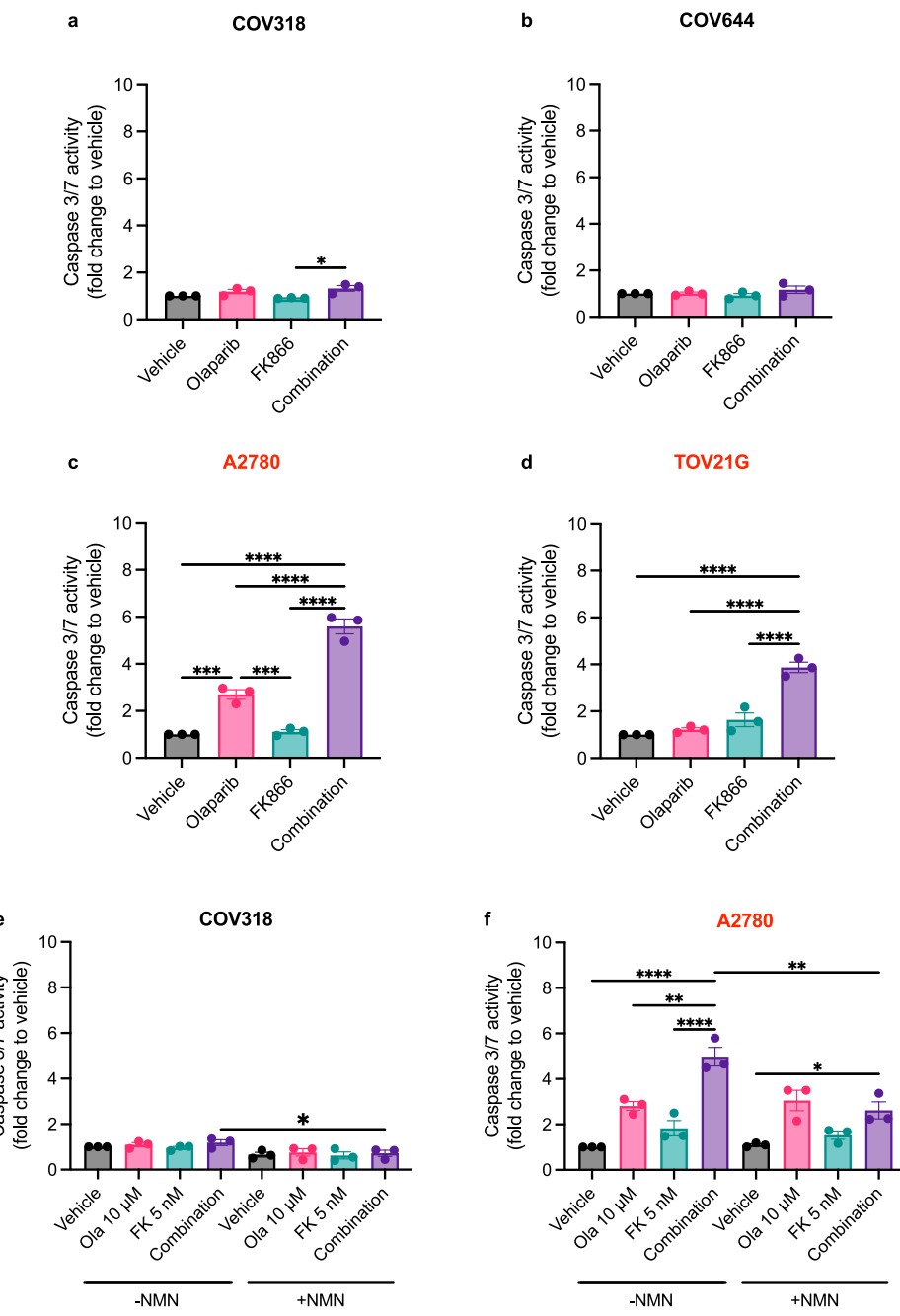

murine model of HGSC with a *Pten* deletion[25] (i.e. the ID8-*Trp53*$^{-/-}$; *Pten*$^{-/-}$model). The combination induced a significant decrease in cell biomass when compared to either monotherapy treatment, which was partially rescued by supplementing medium with NMN (Fig. 7a). Importantly, combination treatment led to significant upregulation of caspase 3/7 activity (Fig. 7b), ROS levels (Fig. 7c) and γH2AX foci formation (Fig. 7d).

We then evaluated the combination in vivo. C57BL/6J mice were injected with ID8-*Trp53*$^{-/-}$; *Pten*$^{-/-}$cells and treated with 10-doses (2-day drug holiday between doses 5–6) of vehicle, FK866 (10 mg/kg), olaparib (50 mg/kg) or the combination. Treatment commenced after the formation of omental tumours (~14 days) (Fig. 8a). 24 h after the final injection, there was a significant decrease in omental tumour weight observed in mice that received the combination (Fig. 8b). NAD$^+$ levels were also significantly reduced following treatment with FK866 or the combination (Fig. 8c). Importantly, no significant reduction in mouse

weight was observed in any condition (Supplementary Fig. 8), and no changes in behaviour or signs of pain were observed in mice, indicating that the combination is tolerable.

Finally, we assessed the effect on survival (Fig. 8d). A similar dosing strategy was used, but mice received 8-doses (3-day drug holiday between doses 4–5) of FK866 (10 mg/kg), olaparib (50 mg/kg), the combination or vehicle. Mice in the vehicle group had a median survival of 31 days, and this only increased to 32 days in mice treated with FK866 (p = NS). Olaparib treatment did provide a moderate survival benefit compared to the vehicle group, increasing median survival to 36 days (p = 0.0006). However, the combination provided the greatest survival benefit, with a median survival of 42.5 days. This was a significant improvement compared to the vehicle (p = 0.0006) and olaparib (p = 0.0007) groups (Fig. 8e). Importantly, there was again not a significant reduction in mouse weight in any condition (Supplementary Fig. 9).

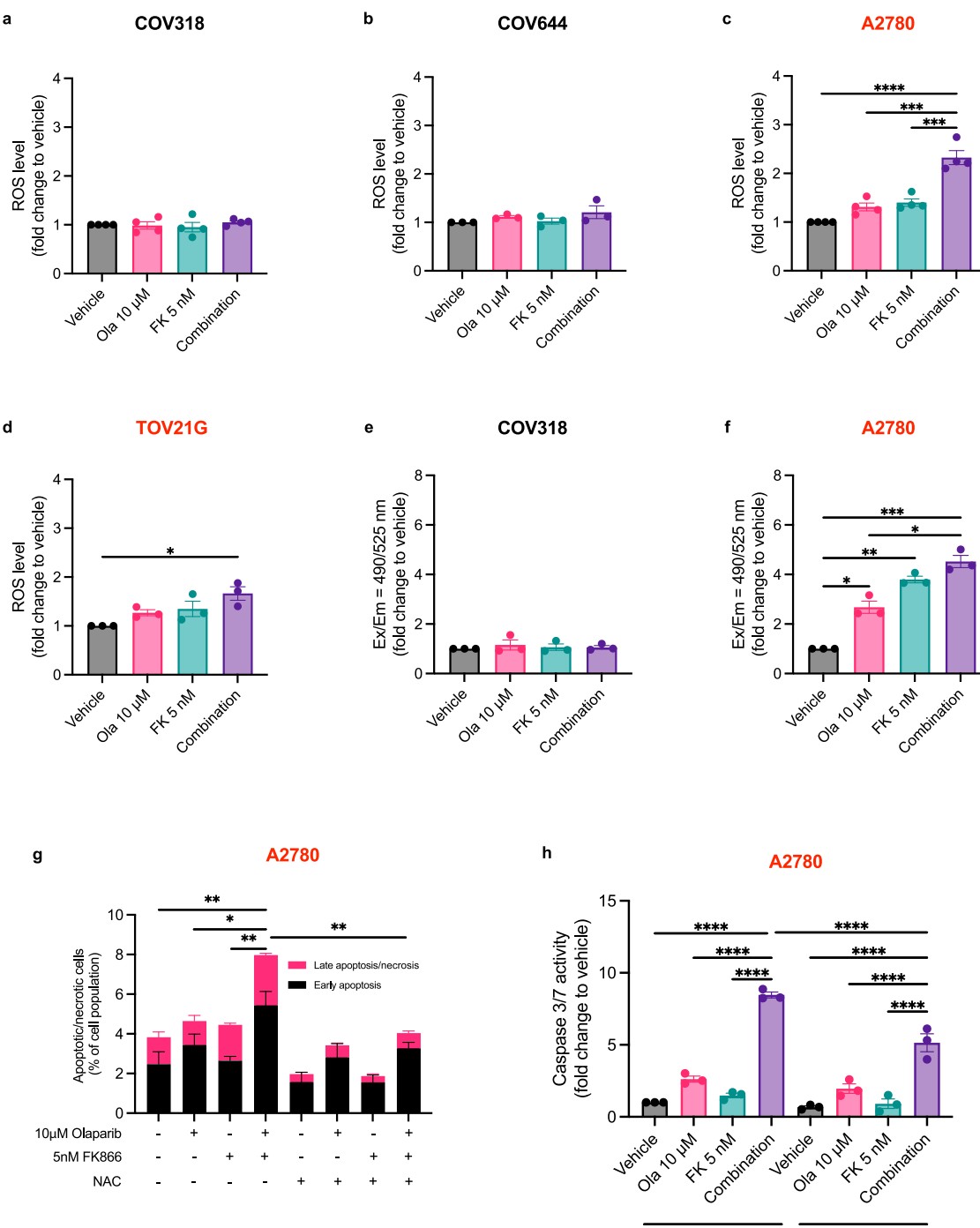

**Fig. 5 | Combination treatment increases ROS generation and mitochondrial dysfunction in RAS/PI3K pathway mutant A2780 cells. a–d** COV318, COV644, A2780 and TOV21G cells were treated for 24-h with the combination and ROS generation was assessed using the ROS-Glo™ H₂O₂ Assay. **e, f** Mitochondrial membrane potential was also assessed after 24-h treatment using the JC-10 assay. Increases indicate mitochondrial membrane depolarisation. **g** A2780 cells were treated for 24-h with vehicle, olaparib, FK866 or the combination before measuring the level of apoptosis/necrosis. As shown in Supplementary Fig. 5a live (Annexin V⁻ and propidium iodide⁻ (PI)), early apoptosis (Annexin V⁺ and PI⁻) and late apoptosis/necrosis (Annexin V⁺ and PI⁺) events were quantified. Data was generated using the Annexin V apoptosis assay. **h** A caspase-rescue experiment was performed in A2780 cells following 48-h treatment with the combination. Data were generated using the Caspase-Glo® 3/7 assay. All data were normalised to cell biomass (SRB assay). **g, h** Where indicated, medium was supplemented with 5–10 mM of the ROS scavenger NAC (see 'Methods' section for further details). Data represent the average ± SD of at least three independent experiments. Statistical significance was determined using two-way ANOVA followed by Tukey's multiple comparisons test (p < 0.05, **p < 0.01, *** p < 0.001, **** p < 0.0001).

## Discussion

A growing body of evidence has emerged demonstrating that combining NAMPT and PARP inhibitors could be a viable strategy to improve the efficacy of PARPis in diseases such as ovarian cancer[9–11]. However, because dose-limiting toxicity has been observed in clinical trials of NAMPTis[12,13],

we set out to identify novel determinants of sensitivity to NAMPTis and the combination.

In this study, we provide evidence that EOC cell lines harbouring RAS/PI3K pathway mutations are sensitive to NAMPT-inhibition with FK866. In these cell lines, lower doses of FK866 are required for a critical reduction

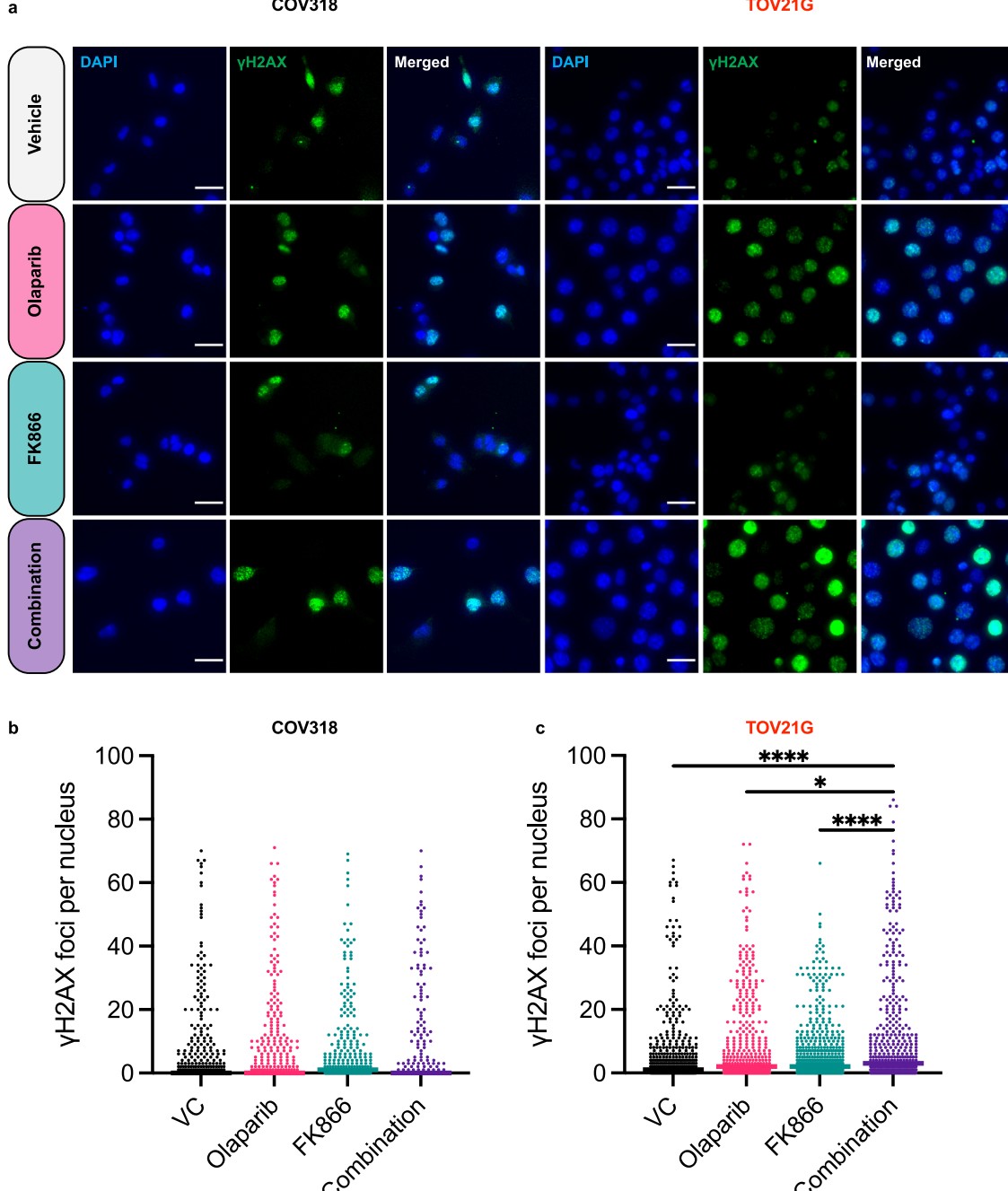

**Fig. 6 | Combined olaparib and FK866 treatment increases γH2AX foci formation in RAS/PI3K pathway mutant TOV21G cells. a** Immunofluorescence images of COV318 and TOV21G cells treated for 24 h with 10 µM olaparib and/or 5 nM of FK866, a 30 µm scale bar is shown. **b, c** Quantitation of γH2AX foci in COV318 and in TOV21G cells. Values derived from 400 cells and the median is shown. Data is representative of at least two independent experiments. Statistical significance was determined using two-way ANOVA followed by Tukey's multiple comparisons test ($p < 0.05$, **$p < 0.01$, *** $p < 0.001$, **** $p < 0.0001$).

in $NAD^+$ and ATP production to be observed (Fig. 2b, d, e), leading to NAMPTi-induced cell death[28,29]. Previous studies have shown *KRAS*-mutant tumours have lower levels of intracellular $NAD^+$[19,20], presumably due to increased metabolic demand. Indeed, using LC-MS/MS-based metabolomics we confirmed that the *KRAS*-mutant OVCAR-8 cell line possesses significantly smaller NMN/$NAD^+$ pools, that are more susceptible to NAMPT-inhibition when compared to the RAS/PI3K-wildtype COV318 cell line (Supplementary Fig. 2a, b). Furthermore, using the $NAD^+$-Glo assay we confirmed RAS/PI3K-mutants had the smallest $NAD^+$ pools in our EOC cell line panel (Fig. 2c). Interestingly, $BRAF^{V600E}$ mutant melanoma models are known to become dependent on the salvage pathway to synthesise $NAD^+$, and are sensitive to NAMPTis[22,23,36,37], suggesting oncogenic

activation of RAS/PI3K-signalling may lead to a NAMPT-dependency in more tumour types. There is evidence that suggests NAMPT-dependency in *RAS*-mutant tumours could be due to regulation of its expression by STAT5 (via the BRAF/ERK pathway)[36,37]. However, the PI3K/AKT-axis may also regulate NAMPT-dependency, as its expression has been shown to inversely correlate with PTEN[21]. This highlights oncogenic activation of Ras-Raf-MEK-ERK and PI3K-AKT-mTOR pathways can lead to the development of an actionable metabolic bottleneck in the $NAD^+$ salvage pathway. However, future studies should characterise how different mutations in these pathways can lead to a NAMPT-dependency.

The combination of FK866 and olaparib has previously been shown to be synergistic in ovarian cancer, including in models that are intrinsically

**Fig. 7 | The combination has activity in ID8 *Trp53*[−/−]; *Pten*[−/−] cells in vitro. a** ID8 cells were co-treated with olaparib and FK866 (±250 μM NMN) for 3-days. Cell biomass was measured using the SRB assay. The influence of the combination (10 μM olaparib and 5 nM FK866) on **b** caspase 3/7 activity (48-h), **c** ROS-generation (48-h) and **d** γH2AX foci formation (24-h) was also assessed. Values are derived from 400 cells and the median is shown, data are representative of three independent experiments. **a–c** Data represent the average ± SD of three independent experiments. Statistical significance was determined using two-way ANOVA followed by Tukey's multiple comparisons test (p < 0.05, **p < 0.01, *** p < 0.001, **** p < 0.0001).

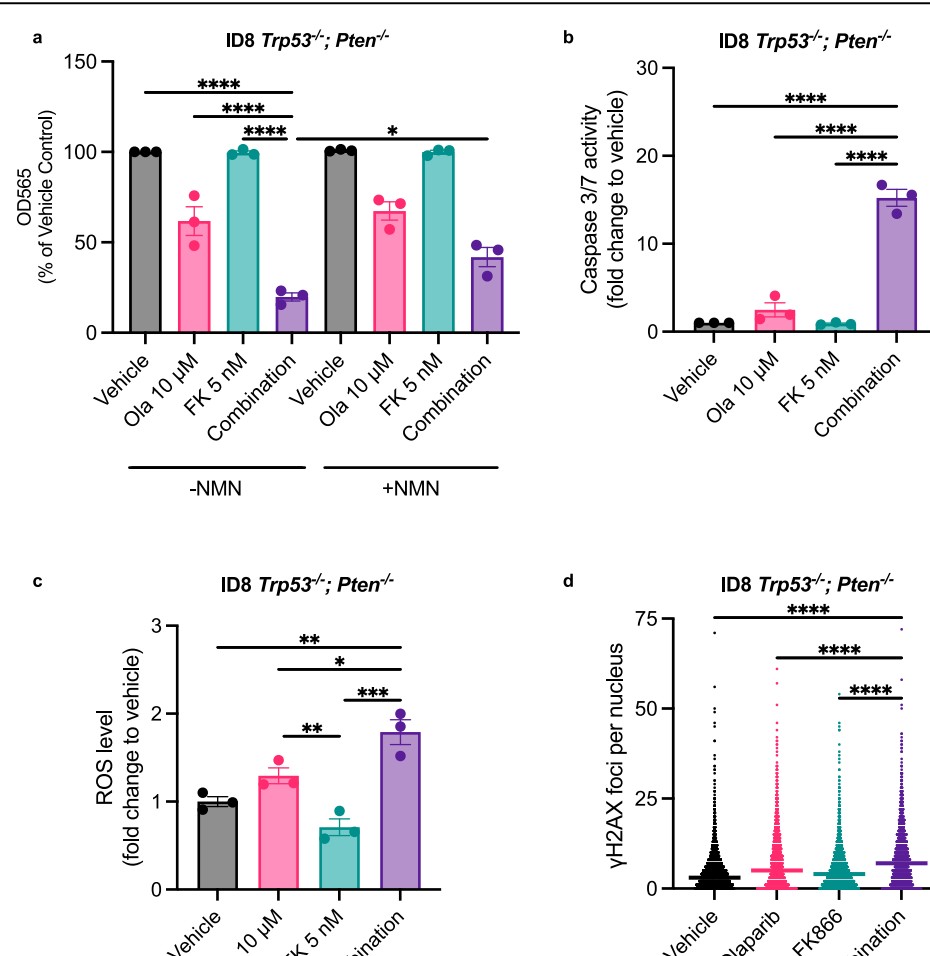

resistant to PARPis[11], as was also observed in our 2D combination experiments (Fig. 3a and Supplementary Fig. 4). However, when assessing the combinations influence on cell viability in 3D (Fig. 3b–e) or on apoptosis (Fig. 4a–d and Supplementary Fig. 6) we demonstrate that RAS/PI3K-mutant EOC cell lines benefit to a greater extent following treatment with the combination. Importantly, we confirmed that synergy is dependent on NMN/NAD[+] depletion (Fig. 4e, f), which is in line with previous observations[9–11].

Previous studies have demonstrated that the on-target activity of FK866 activates apoptotic signalling by inducing mitochondrial dysfunction. Maintenance of MMP is essential to maintain the election transport chain to generate ATP. NAD[+] plays an important role in this process, its depletion (and NADH consequently) will limit the mitochondrial respiratory substrates availability, leading to a loss of MMP[28,29]. Indeed, our data demonstrate that in the RAS/PI3K-mutant A2780 cell line, FK866 treatment leads to a loss of MMP (Fig. 5f). To our surprise, given that PARP inhibition is reported to lead to mitochondrial protection[38], we did observe a significant loss of MMP compared to the vehicle group following olaparib treatment (Fig. 5f). Furthermore, combined olaparib and FK866 treatment led to the largest MMP loss, although this was not statistically significant when compared to FK866-monotherapy. Generation of ROS is another indicator of defective electron transport in mitochondria, and FK866-induced NAD[+] depletion has previously been shown to increase ROS levels[28,29]. However, we only observed a significant increase in ROS levels in the RAS/PI3K-mutant A2780 and TOV21G cells following treatment with the combination (Fig. 5c, d). Importantly, using the ROS-scavenger NAC[32], we demonstrated that it is possible to partially rescue apoptosis following combination treatment in A2780 cells (Fig. 5g, h), confirming increased

ROS generation upregulates apoptosis. Taken together, these data provide evidence that ROS/MMP levels can be altered following combination treatment and contribute towards the induction of apoptosis, further extending our understanding of the mechanism of action of this combination.

The combination has previously been shown to upregulate DNA damage (i.e. increased γH2AX foci)[9–11], and encouragingly we provide evidence that γH2AX foci formation is increased in RAS/PI3K-mutant TOV21G cells (Fig. 6). Treatment with a PARPi, such as olaparib, is also associated with marked G2/M cell cycle arrest[34]. Furthermore, treatment with FK866 has also been shown to cause a G2/M-phase arrest-like state[23]. However, to the best of our knowledge the influence of the combination on the cell cycle has not been investigated. Interestingly, in RAS/PI3K-mutant A2780 and TOV21G cells, we observed a significant increase in the proportion of cells in the G2/M-phase following combination treatment, suggesting it enhances G2/M-phase arrest (Supplementary Fig. 7). Given a substantial amount of mitochondrial energy (e.g. ATP) is required for cell-cycle progression, especially at the G2/M phase[23,39], it is possible the combination blocks cells in the G2/M phase since mitochondrial energy production is severely impaired (e.g. ATP loss, MMP loss and increased ROS) in RAS/PI3K pathway mutant cells.

Our combination experiments in ID8 *Trp53*[−/−]; *Pten*[−/−] cells confirmed that similar trends were observed, namely, growth inhibition that could be rescued by NMN (Fig. 7a), and a significant increase in apoptosis (Fig. 7b), ROS (Fig. 7c) and γH2AX foci (Fig. 7d) could be observed when compared to monotherapy groups. Finally, our in vivo data demonstrate the efficacy of combined olaparib and FK866 treatment in inhibiting the growth of a *Pten*-null tumour (Fig. 8).

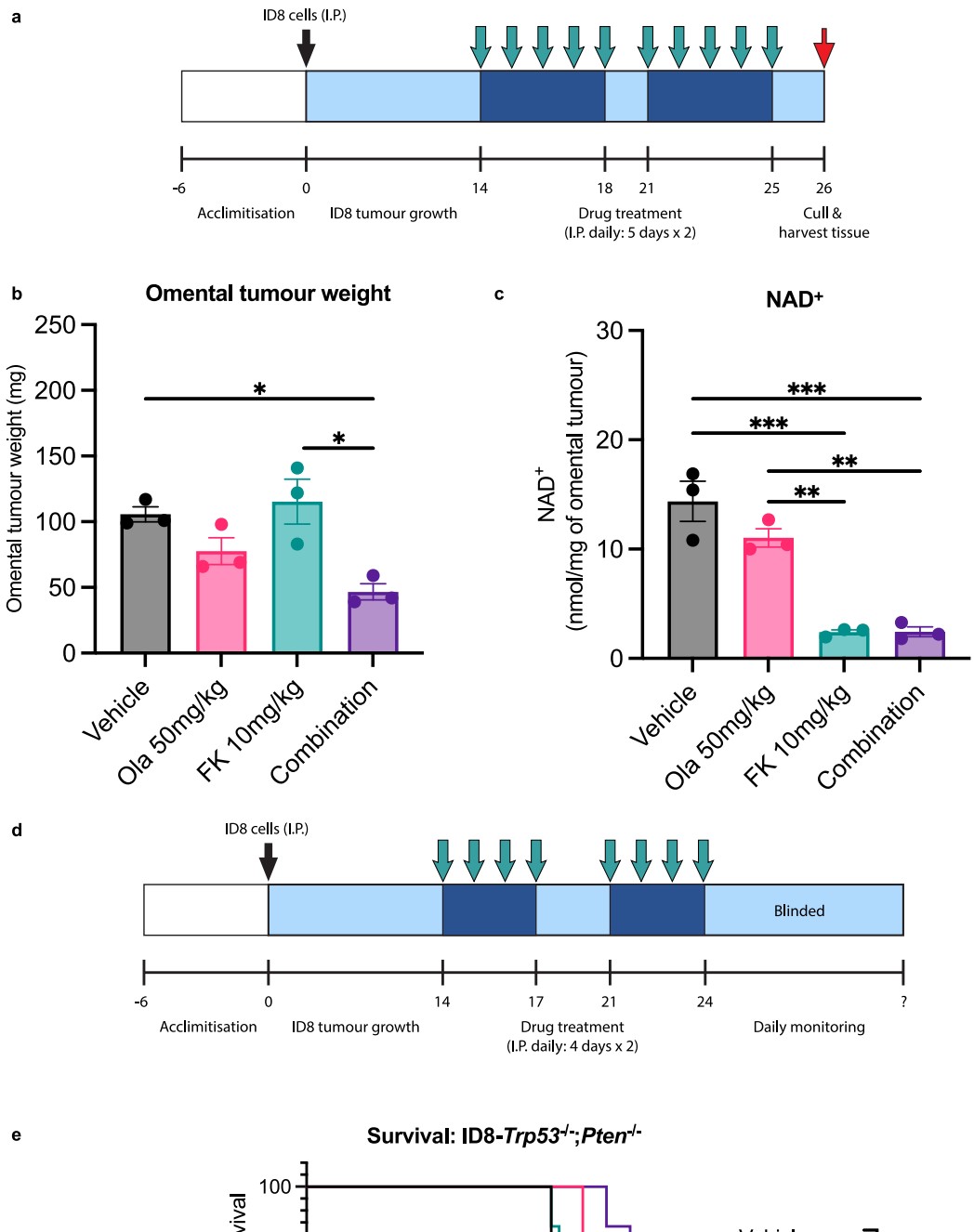

**Fig. 8 | Combined olaparib and FK866 treatment reduces omental tumour weight and improves survival outcomes in the ID8-*Trp53*$^{-/-}$; *Pten*$^{-/-}$ model. a** For the endpoint experiment ID8-*Trp53*$^{-/-}$; *Pten*$^{-/-}$ cells were injected intraperitoneally (I.P.) into C57BL/6J mice and allowed to grow for 14-days before mice were treated with vehicle, olaparib, FK866 or the combination. Twenty four hours after the final drug treatment, omental tumours were **b** weighed and **c** harvested for NAD$^+$ quantification by LC-MS/MS. Data represent the average ± SD of three independent tumours. Statistical significance was determined using two-way ANOVA followed by Tukey's multiple comparisons test ($p < 0.05$, **$p < 0.01$, *** $p < 0.001$, **** $p < 0.0001$). **d** For the survival experiment mice, were randomised into different groups for drug treatment. Researchers were blinded after the final drug treatment until the study's completion for unbiased determination of humane endpoint. **e** For Kaplan–Meier curves, the log-rank (Mantel–Cox) test was used to determine statistical differences between survival curves. Bonferroni correction was used for the multiple comparisons test ($p < 0.05$, **$p < 0.01$, *** $p < 0.001$, **** $p < 0.0001$).

Based on observations from this study and others[9–11], we believe the synthetic lethality observed between a PARPi and NAMPTi combination is likely to be due to a mechanism previously proposed by Bajramin and colleagues[9]: (i) PARP uses $NAD^+$ as a substrate, treatment with a PARPi reduces PARylation leading to PARP trapping[40], (ii) competition between $NAD^+$ and olaparib for binding to the catalytic domain of PARP1/2 occurs[41] but reducing $NAD^+$ levels with a NAMPTi increases the binding of olaparib to PARP1/2. (iii) This results in the formation of persistent DNA lesions (i.e. γH2AX foci) and impaired DNA repair, exacerbating the deleterious effects of PARP inhibitors on cells (e.g. increased apoptosis). Considering this, we propose that because RAS/PI3K-mutant EOC cell lines possess smaller $NAD^+$ pools (Fig. 2c and Supplementary Fig. 2b) that are more readily depleted by the NAMPTi FK866 (Fig. 2d and Supplementary Fig. 2b), lower doses of the PARPi/NAMPTi combination are required to observe synergy (Supplementary Fig. 4). Importantly, RAS/PI3K-mutations occur in each subtype of EOC (e.g. *BRAF*, *KRAS*, *NRAS*, *PIK3CA* mutations or *NF1* and *PTEN* loss)[16,17,42]. In the most common EOC subtype (i.e. HGSC), the RAS/PI3K pathway is dysregulated in ~45% of cases through pathway mutations and copy number alterations[16], this could substantially expand the range of genetic backgrounds for which the combination is relevant.

In future, it would be interesting to investigate whether targeting other parts of the $NAD^+$ biosynthesis pathway could sensitise RAS/PI3K-mutants to PARPis. For example, the Preiss–Handler pathway could be targeted by inhibiting nicotinate phosphoribosyltransferase (NAPRT)[43]. Interestingly, the PARPi/NAMPTi combination has recently been shown to be synergistic in *NAPRT*-silenced renal cell carcinoma[44]. It could also be interesting to investigate whether the therapeutic window could be further maximised through the use of antibody-drug conjugates (ADC) to improve the delivery of NAMPTis to cancer cells, whilst sparing healthy tissue[45]. Encouragingly, NAMPTi-ADCs have shown potent anti-cancer activity in vivo[46–48].

One limitation of our study is that ID8 cells derive the ovarian surface epithelium rather than the fallopian tube, which is the origin of most HGSC cases[49,50]. Furthermore, it would be advantageous to validate our findings in genetically engineered murine models that better represent other ovarian cancer subtypes[51,52], and also in patient-derived models. It should also be noted, the A2780 cell line has previously been misclassified as HGSC, but recent reports demonstrate it is a model of endometroid ovarian cancer, as originally described[26,53].

Nonetheless, taken together, our data suggest that RAS/PI3K pathway mutations sensitise EOC cells to NAMPT inhibition, increase the therapeutic window of the PARPi/NAMPTi-combination, and could extend the benefit of PARPis to a broader range of EOC patients (e.g. those without a *BRCA1/2* mutation). Given RAS/PI3K pathway mutations are observed in each EOC subtype and are associated with poorer PARPi responses, we suggest that combining PARPis and NAMPTis may be a promising strategy in this group of EOC patients. Especially with the development of new generations of NAMPTi and PARPi, such as OT-82[33,54–56] (currently in clinical trials: NCT03921879) and Saruparib[57,58] (clinical trial: NCT04644068), with more favourable toxicological profiles, further expanding the possibilities for PARPi/NAMPTi combination therapy.

## Methods

### Cell culture
The following human cell lines were used in the present study: COV318 (ECACC, #07071903), COV504 (ECACC, #07071902), COV644 (ECACC, #07071908), KURAMOCHI (acquired from Dr. Haonan Lu, Imperial College London), OVSAHO (JCRB, #JCRB1046) 59 M (ECACC, #89081802), A2780 (ECACC, #93112519), HEY-A8 (acquired from the characterised Cell Line Core Facility at MD Anderson Cancer Centre), TOV21G (ATCC, #CRL-3577) and OVCAR-8 (acquired from the National Cancer Institute). All human cell lines were cultured in phenol red-free DMEM (ThermoFisher, A1443001) supplemented with 5.6 mM glucose (ThermoFisher, A2494001), 10% foetal bovine serum (FBS; ThermoFisher, 10270106), 2 mM L-glutamine (ThermoFisher, 25030-081), 100 units/mL penicillin, 100 μg/mL streptomycin (ThermoFisher, 15070-063). Cells were

maintained at 37 °C in a humidified atmosphere containing 10% $CO_2$. Cell lines were authenticated using Short Tandem Repeat (STR) profiling by Public Health England. To avoid genetic drift human cell lines were cultured for no more than ten passages. Cell lines were regularly tested for mycoplasma contamination.

The generation of ID8-*Trp53*⁻/⁻; *Pten*⁻/⁻ cells has been described previously[25]. For in vitro experiments, these cells were adapted to physiological glucose (5.6 mM) over the course of three passages. After adaptation, cells were cultured in phenol red-free DMEM supplemented with 5.6 mM glucose, 10% foetal bovine serum, 2 mM L-glutamine, 100 units/mL penicillin, 100 μg/mL streptomycin and 1 mM sodium pyruvate. For in vivo experiments, cells were cultured as described previously[24,25], in phenol red-free DMEM supplemented with 25 mM glucose, 4% FBS, 2mM L-glutamine, 100 units/mL penicillin, 100 μg/mL streptomycin, 1 mM sodium pyruvate (ThermoFisher, 11360070) and 1× Insulin-Transferrin-Selenium (ThermoFisher, 41400045).

### Annexin V and PI assay
COV318, A2780 and TOV21G cells were seeded into 6-well plates and allowed to attach overnight. The next day, medium was replaced with DMEM supplemented with 0.1% DMSO, FK866 and/or olaparib. After 24-h, cell surface annexin V expression was quantified as described previously[59]. Live cells were detached from plates and combined with floating cells. Cells were washed twice with cold PBS, resuspended in Annexin V binding buffer (0.01 M HEPES, 0.14 M NaCl, 2.5 mM $CaCl_2$) and single-cell suspensions were produced using a 30 μm cell strainer. Cells were stained using a solution that contained Cy5.5-conjugated Annexin V (2.5 μL per test, BD Biosciences, #559925) and PI solution (2.5 μL per test, Invitrogen, #00-6990-50). Cells were then incubated for 15-min in the dark and subsequently analysed using a BD FACS Canto analyser and data were analysed using FlowJo V10.6.2. Positive controls for each stain were included in each repeat for colour compensation.

### ATP assay
Cell lines were seeded onto two parallel plates at a density of 10,000 cells per well. 24-h after seeding cells were treated with FK866. After 48-h treatment, SRB plates were processed using the SRB assay protocol, and white walled plates were processed using the CellTiter-Glo® assay, following manufacturer's guidelines. Luminescence was measured on a FLUOstar Omega Microplate reader (BMG LABTECH). Cell-free wells were used as blanks to remove background luminescence. ATP were normalised to cell biomass (SRB assay).

### Bioinformatics
Cell sensitivity data ($IC_{50}$) to FK866 (Daporinad) were downloaded from the Genomics of Drug Sensitivity in Cancer web portal (www.cancerrxgene.org, GDSC1). Mutation status was based on information available from the CCLE[26].

### Caspase-Glo® 3/7 assay
Twenty four hours after seeding, cells were treated with vehicle, olaparib and/or FK866. After 2-day treatment, SRB plates were processed using the SRB assay protocol, and white-walled plates were processed using the Caspase-Glo® 3/7 Assay, following the manufacturer's guidelines. Luminescence was measured on a FLUOstar CLARIOstar Plus Microplate Reader (BMG LABTECH). Cell-free wells were used as blanks to remove background luminescence. Caspase data were normalised to cell biomass. Data was then normalised to VC wells.

### Cell cycle assay
COV318, A2780 and TOV21G cells were seeded into 6-well plates and allowed to attach overnight. The next day, medium was replaced with DMEM supplemented with 0.1% DMSO, FK866 and/or olaparib. Cell cycle assay was performed as described previously[59]. Briefly, after 24-h drug treatment, cells were incubated with 200 mM IdU (Sigma, I7125) and

washed with PBS twice before collection by trypsinization (Gibco, 12604013). After fixing overnight at 4 °C in 70% ethanol, chromosomes were denatured by adding 2 M HCl with 0.5% Triton X-100 (Sigma-Aldrich, T8787) with agitation and incubating for 30-min. Next, 0.1 M of NaB4O7 (pH 8.5; Sigma, B9876) was added to samples, and they were incubated for 5-min to neutralise the HCl. The samples were then incubated with 1% BSA (Sigma, A2153) and 0.2% Tween 20 (Sigma, P1379) in PBS for 30-min for blocking. To probe IdU, the samples were incubated with mouse anti-IdU antibody (Abcam, ab6326; 1:25 in blocking buffer (BD Bioscience, 347580)) for 60-min each at room temperature. The samples were then washed with blocking buffer (BD Bioscience, 347580) twice and incubated with AF488-conjugated goat anti-mouse antibody (Invitrogen, A11001; 1:100 in blocking buffer (BD Bioscience, 347580)) before 15-min incubation with the FxCycle PI/RNase staining solution (Invitrogen, F10797) in the dark. The samples were then analysed using BD FACS Canto analyser and data were analysed using FlowJo V10.6.2.

## Immunofluorescence

For immunofluorescence assays, COV318, TOV21G and ID8-$Trp53^{-/-}$; $Pten^{-/-}$ cells were treated for 24-h with vehicle, FK866 and/or olaparib. At endpoint, cells were washed with 100 μL PBS before being fixed for 15 min (RT) with 50 μL 4% PFA (ThermoFisher, 11586711) in PBS. Plates were washed with 100 μL PBS and permeabilised for 20 min with 50 μL 0.1% Triton-X100 (Sigma–Aldrich, T8787) in PBS. After two PBS washes, a blocking step was performed (1% BSA and 2% FBS in PBS) for 30 min at RT. 1% BSA (in PBS) alone or with the γH2AX (1:500) antibody (Sigma-Aldrich, 05-636) was added and plates were incubated overnight at 4 °C. The next day, wells were washed three times with PBS and once with 1X TBS-T. Next, cells were incubated in the dark (RT) for 2 h with 50 μL 1% BSA (in PBS) staining solution containing: Goat anti-Mouse IgG (H + L) Cross-Adsorbed Secondary Antibody, Alexa Fluor™ 488 (1:500, Invitrogen, #A11001) and 1 μg/mL DAPI (Sigma-Aldrich, #9542). Wells were then washed three times with PBS and one time with 1X TBS-T, before adding 200 μL of PBS to each well. Prior to imaging AbsorbMax™ film (Sigma-Aldrich, Z722537) was placed on top of 96-well plates to minimise background fluorescence. Images were then taken on an IN Cell™ Analyser 1000 (GE Healthcare Life Sciences) and the accompanying INCell analyser 3.5 software. A 20× objective lens was used and 25 fields were captured in each well. CellProfiler V4.0.6 (Broad Institute of MIT and Harvard) was used for image analysis.

## In vivo experiments

In vivo experiments complied with UK welfare guidelines and were conducted under UK Home Office personal and project licence (PA780D61A) authority in dedicated facilities. All experiments were approved by the Imperial College Animal Welfare and Ethics Review Board (reference IM170119) and protocols were designed before experiments commenced. We have complied with all relevant ethical regulations for animal use. $5 \times 10^6$ ID8-$Trp53^{-/-}$; $Pten^{-/-}$ cells were injected into 6–8-week-old female C57BL/6 mice by intraperitoneal (I.P.) injection. Mice were monitored daily by observation of weight and overall health. Vehicle solution consisted of 3.5% DMSO and 6.5% Tween 80 (Sigma-Aldrich, P4780) in 0.9% saline. The vehicle solution was used to prepare suspensions of olaparib (50 mg/kg) and FK866 (10 mg/kg). Each suspension was vigorously vortexed and sonicated to break down particulates, and drug solutions were inspected using a haemocytometer to ensure no large particles remained. Drugs were stored as aliquots at −20 °C and thoroughly vortexed prior to use.

Vehicle, olaparib and/or FK866 were administered as 200 μl IP injections. For the endpoint experiment, mice received daily I.P. injections on days (D) 14-18 and D21-25 inclusive. Three mice were included per treatment group. The day after the final injection (D26) mice were killed. Organs/tumour were collected, weighed and snap frozen.

For the survival experiment, mice received daily I.P. injections on D14-17, and D21-24 inclusive. Six mice were included per treatment group. After completion of the dosing schedule, researchers were blinded until the study's completion for unbiased determination of humane endpoint. Mice were killed upon reaching a humane endpoint. Organs/tumour were collected, weighed and snap frozen.

The ID8 model produces disseminated intraperitoneal disease. It does not form tumours that can be measured using callipers, so maximal tumour measurements/volumes could not be taken. All in vivo work was performed at the Central Biological Services facility, Imperial College London in accordance with the U.K. Animals (Scientific Procedures) Act 1986 under Project License PA780D61A and following approval by the Imperial College AWERB (Animal Welfare and Ethical Review Body) (i.e. the Institutional Animal Care and Use Committee). Mice were monitored regularly and killed upon reaching the moderate severity limit as permitted by the Project License limits, which included weight loss, reduced movement, hunching, jaundice, and abdominal swelling. These limits were not exceeded in any experiments.

## MMP assay

COV318, A2780 and TOV21G were seeded in triplicates into black 96-well plates. The cells were also seeded into parallel clear 96-well plates at the same density for normalisation against cellularity. After overnight incubation to allow attachment to the plates, the medium was replaced with fresh DMEM with 0.1% DMSO, FK866 and/or Olaparib. After 48-h drug treatment, mitochondrial membrane potential (MMP) was measured using the JC-10 assay kit (Abcam, ab112134) according to the manufacturer's instructions. In brief, 50 μl of assay buffer A with JC-10 dye was added to each well and incubated at room temperature for 1-h. Next, 50 μl of assay buffer B was added to each well to lyse the cells and fluorescence at excitation/emission = 490/525 nm (cut off at 515 nm) was measured by FLUOstar CLARIOstar Plus Microplate Reader (BMG LABTECH). The parallel clear plates were processed according to the SRB assay. Cell-free wells were used for background subtraction. Fluorescent intensity was normalised against SRB data for correction of cell biomass.

## NAD⁺/NADH-Glo™ assay

Cell lines were seeded onto two parallel plates. A clear plastic plate was used for an SRB assay and white walled 96-well plates were used for the NAD/NADH-Glo™ assay (Promega, G9072). Twenty four hours after seeding, cells were treated with FK866. After 24-h treatment, SRB plates were processed using the SRB assay protocol, and white walled plates were processed using the NAD⁺/NADH-Glo™ assay. In brief, medium was aspirated from wells and replaced with 50 μL PBS, or 50 μL PBS with diluted NAD+ standards (50–2000 nM) was added into cell free wells. 50 μL 1% dodecyltrimethylammonium bromide (DTAB) with 0.2 N NaOH was added to each well and plates were briefly mixed to ensure homogeneity and cell lysis. Fifty microliters lysate was transferred into an empty well and 25 μL 0.4 N HCl was added. Plates were covered with foil and incubated for 15 min on a ThermoMixer® C (Eppendorf) heat block at 60 °C. Plates were allowed to equilibrate for 10 min at RT before adding 25 μL 0.5 M Trizma® Base (Sigma-Aldrich, 93352) solution. Fifty microliters of this mixture was then transferred into a white walled 96-well plate along with 50 μL of the NAD⁺/NADH-Glo detection reagent, which was made up with reagent at the following ratio: 1000 μL Luciferin Detection Reagent, 5 μL Reductase enzyme, 5 μL Reductase substrate, 5 μL NAD Cycling Enzyme and 25 μL NAD Cycling Substrate. Plates were mixed for 30 min at RT before luminescence was measured on a FLUOstar Omega Microplate reader (BMG LABTECH). Cell-free wells were used as blanks to remove background luminescence. Data were normalised to VC wells. NAD⁺ levels were determined by normalising to cell biomass (SRB assay).

## ROS assay

COV318, A2780, TOV21G and ID8 $Trp53^{-/-}$; $Pten^{-/-}$ cells were seeded into white 96-well plates, and a parallel clear 96-well plate was seeded for normalisation against cell biomass (SRB assay). After overnight incubation and cell attachment to the plates, medium was removed and 80 μl of fresh DMEM media with 0.1% DMSO, FK866 and/or Olaparib was added to each

well. During ROS rescue experiments, 2-h prior to drug treatment cells were pre-treated with 10 mM NAC. The NAC concentration was reduced to 5 mM during drug treatment. After 24- (human) or 48-h (ID8), $H_2O_2$ was measured using ROS-Glo $H_2O_2$ assay (Promega, G8820) according to the manufacturer's manual. Briefly, 6-h before the endpoint is reached, 20 μL of combined $H_2O_2$ substrate (provided in the kit) and test compound were added to reach the final concentration of $H_2O_2$ substrate at 25 μM and drug concentrations were kept unchanged. After a 6-h incubation, 100 μL ROS-Glo™ detection solution (provided in the kit) was added, and the plates were incubated at room temperature for 20-min. Next, luminescence was measured using a FLUOstar CLARIOstar Plus Microplate Reader (BMG LABTECH). To subtract background signals, cell-free wells were processed using the same protocol. To correct for cell biomass, the luminescence signal was normalised to SRB readings from the parallel plates (SRB assay).

## Sulforhodamine B (SRB) cell growth assay

Human cell lines were seeded on 96-well plates at a pre-determined density. The following day, wells were aspirated and replaced with medium containing vehicle (0.1% DMSO), FK866 (50 pM-50 nM) (APExBIO, A4381) or olaparib (1 nM-200 μM) (Selleckchem, S1060). Both inhibitors were prepared in DMSO (Sigma–Aldrich, D2650). Where indicated, drugs were added in combination and/or medium was supplemented with 250 μM of NMN (Sigma–Aldrich, N3501). After 3- (ID8) or 6-days (Human) of treatment, cells were fixed for 1 h at 4 °C with 10% (w/v) trichloroacetic acid (TCA) solution (Sigma–Aldrich, T6399). Wells were washed with water and left to dry at room temperature (RT) overnight. Cells were stained for 30 min at RT with 0.4% SRB (Sigma–Aldrich, 230162) solution (w/v) in 1% (v/v) acetic acid (ThermoFisher, 10005920). Wells were washed four times with 1% acetic acid (v/v) to remove excess dye and plates were left to dry overnight at RT. Two hundred microliters 10 mM Tris Base solution (Sigma–Aldrich, T1699) was added to wells and plates were placed on a shaker to solubilise the protein-bound SRB dye. Absorbance was then measured on a SpectraMax Gemini XPS Microplate Reader (Molecular Devices) at 565 nm. A minimum of three technical replicates were used for each condition. Cell-free wells were used as blanks to remove background absorbance. Data were normalised to vehicle control (VC) wells and dose response curves were generated using nonlinear regression function [inhibitor] vs. response (three parameters) in the GraphPad Prism 10.0.0 software. Synergy scores were calculated using SynergyFinder[60].

## Statistics and reproducibility

Unless otherwise stated, statistical tests were performed using GraphPad Prism (10.0.0). Statistical tests used and the number of replicates are described in each figure legend. All in vitro experiments were conducted in at least three independent experiments. Statistical significances are given by the $p$-value: ***$p < 0.001$; **$p < 0.01$; *$p < 0.05$; n.s. (not significant), $p \geq 0.05$. To compare statistical significance, a two-way ANOVA followed by Tukey's multiple comparisons test was performed. Otherwise, statistical significance was determined using the brown-forsythe and Welch ANOVA tests, with Dunnett's t3 multiple comparisons test. Linear regression was performed with GraphPad Prism, and correlation was calculated with the Pearson correlation test. For Kaplan–Meier curves, the log-rank (Mantel-Cox) test was used to determine statistical differences between survival curves. Bonferroni correction was used for the multiple comparisons test.

## Targeted LC-MS/MS analysis for NMN and NAD$^+$ quantification

**Cell lines**. COV318 and OVCAR-8 cells were seeded on 6-well plates at a density of 200,000 cells per well. The following day, the media was aspirated and replaced with fresh media containing vehicle, FK866 and/or olaparib. Separate plates were included for media blanks, extraction blanks and for cell counts at the time of extraction to normalise data by cell number. Twenty four hours after treatment commenced, cell counts were performed on a Vi-CELL XR Cell Viability Analyzer (Beckman coulter). The remaining plates were transferred onto a ThermalTray™ (Biocision) that had been pre-cooled with dry ice and 2 mL Eppendorf

tubes were pre-cooled in a CoolRack®. Wells were aspirated and cells were washed with ice-cold Ringers solution (2 mL, one tablet per 500 mL of MilliQ $H_2O$, Sigma–Aldrich, 96724). Wells were aspirated and cells were quenched with 1 mL of ice cold LC-MS grade 80:20 methanol:water (v/v) (ThermoFisher, 34966) for 20 min. Wells were scraped and the extracts were transferred to the 2 mL tubes. The previous step was repeated with 500 μL 100% methanol. Next, extracts were centrifuged for 20 min at $18,000 \times g$ (4 °C) using a Eppendorf™ Centrifuge 5430R (Eppendorf), to pellet out cellular debris. Supernatants were collected in high recovery LCMS vials, dried under nitrogen flow, and stored at −80 °C. Before LC-MS/MS analysis samples were reconstituted in 50 μL of LC-MS grade $H_2O$ (ThermoFisher, 10505904), vortexed for 1 min and stored in an autosampler at 10 °C.

LC-MS/MS assays were performed using an Agilent 1290 Infinity LC system (Agilent Technologies) and a 4000 QTRAP triple quadrupole instrument (AB SCIEX) coupled to an ESI source (Turbo V). An ACQUITY UPLC® HSS (High Strength Silica) T3 Column (Waters) was used for Reversed-Phase (RP) Liquid Chromatography, with a column temperature of 40 °C. To increase the columns lifetime a ACQUITY UPLC HSS T3 VanGuard Pre-column (Waters Corp, 186003976) was used. Elution was performed at a flow rate of 600 μL/min using the gradient described in Supplementary Table 1 and 5 μL of sample was used per injection. Needle wash steps were performed with Acetonitrile:Isopropanol:$H_2O$ (40:40:20, v/v/v) (ThermoFisher, 10055454) (ThermoFisher, 10684355). Mobile phase A was comprised of $H_2O$ and 0.2% formic acid (v/v) (Scientific Laboratory Supplies, 56302). Mobile phase B contained acetonitrile (ACN) and 0.2% formic acid (v/v).

To optimise conditions for targeted metabolomics in Multiple Reaction Monitoring (MRM) mode (QqQ) standards for each compound were directly infused into the MS/MS analyser. From this MRM transitions, declustering potential (DP) and collision energy (CE) values were identified for each target analyte as shown in Supplementary Table 2.

The Analyst (v1.6.2, SCIEX) software was used to perform peak integration. Purified stock compounds were used to produce a calibration curve (0.025-10 μg/mL) for each run. Using this and cell counts, molar concentrations per million cells were calculated for each metabolite.

**Omental tumours**. Roughly 10–15 mg of omental tumour was weighed and placed into clean homogeniser tubes containing 0.8 mL ice cold 80% Methanol and 0.1 mm Precellys® Glass Beads (ThermoFisher, 12927835). Tissue was homogenised using the Precellys Evolution tissue homogeniser (Bertin) with an activated Cryolys Evolution cooling system ($18,000 \times g$, $4 \times 20$ s cycle with a 30 s pause, two cycles). Tubes were centrifuged at $21,000 \times g$ for 10 min (4 °C). Supernatants were collected in Eppendorf tubes. An additional 800 μL 80% Methanol was added, the previous steps were repeated, and extracted supernatants for each sample were pooled. Samples were then dried under nitrogen flow and stored at −80 °C. A processing blank was also included during the extraction. Samples were then subjected to dual-phase extraction. Three hundred microliters chloroform (ThermoFisher, 390760025)/methanol (2:1) was added to each sample, and tubes were vortexed. Tubes were vortexed and subsequently centrifuged at $21,000 \times g$ for 10 min (4 °C). Each sample's aqueous (upper) and organic (lower) layers were transferred into separate LC-MS vials, before drying under nitrogen flow, and storage at −80 °C.

On the day of LC-MS/MS assays, omental tumour (aqueous phase) samples were re-suspended in 100 μL $H_2O$ and vortexed for 30 s and stored in an autosampler at 10 °C. LC-MS/MS assays were performed using an Agilent 1290 Infinity LC system (Agilent Technologies) and a 4000 QTRAP triple quadrupole instrument (AB SCIEX) coupled to an ESI source (Turbo V).

Acquity UPLC ® BEH Amide 1.7 μm 2.1 × 150 mm Column (Waters, 186004802) was used for HILIC Liquid Chromatography, with a column temperature of 50 °C. To increase the columns lifetime a Acquity UPLC ® BEH Amide 1.7 μm VanGuard™ Pre-Column 2.1 × 5 mm (Waters, 186004799) was used. Elution was performed at a flow rate of 500 μL/min

using the gradient described in Supplementary Table 3 and 5 µL of sample was used per injection. Needle wash steps were performed with Acetonitrile:$H_2O$ (1:1, v/v). For HILIC positive mode, mobile phase A was comprised of ACN and 0.1% formic acid (v/v). Mobile phase B contained $H_2O$, 20 mM ammonium formate (Supelco, 70221) and 0.1% FA. Peak integration was performed using the Analyst (v1.6.2, SCIEX) software. For concentration determination, molar concentrations were calculated per mg of omental tumour tissue. MRM transitions, declustering potential (DP) and collision energy (CE) values were identified for each target analyte as shown in Supplementary Table 4.

## Western blotting

Total proteins were extracted from cultured cells using RIPA lysis buffer (Sigma-Aldrich, R0278) supplemented with a Halt™ Protease and Phosphatase Inhibitor Cocktail (ThermoFisher, 78440). To facilitate protein extraction lysates were vortexed for 30 s and then placed on ice for 10 min. This was repeated two more times. Samples were centrifuged at $17,400 \times g$ for 10 min (4 °C) and protein supernatants were collected. Protein concentrations were quantified using a Pierce™ BCA Protein Assay Kit (ThermoFisher, 23225).

30µg of protein was loaded onto 4–20% Mini-PROTEAN® TGX™ Precast Protein Gels (BIO-RAD, 4561094 or 4561096) for electrophoresis (200 V for 50 min), and then a wet transfer (100 V for 60 min) onto a nitrocellulose membrane (BIO-RAD, 1620115) was performed. Membranes were blocked for 1 h (RT) in 5% milk (Mercy, 70166) (total protein) or 5% BSA (Sigma–Aldrich, SRE0096) (phospho protein) in tris-buffered saline containing 0.1% of Tween 20 (TBST). Next, membranes were incubated overnight at 4 °C with primary antibodies in milk (total) or BSA (phospho). Membranes were washed three times with TBST and incubated for 5 min at RT with Supersignal™ West Pico plus chemiluminescent substrates (ThermoFisher, 34580). Blots were developed using Amersham Hyperfilm ECL film (GE Healthcare, 28-9068).

The following primary antibodies were used in this study: mouse anti-β-actin (1:10,000, Abcam, #Ab6276), rabbit anti-AKT (pan) (C67E7) (1:2500, Cell Signalling, #4691), rabbit, anti-phosphor-AKT (Ser473) (1:2000, Cell Signalling, #4060), rabbit anti-p44/42 MAPK (Erk1/2) (1:2000, Cell Signalling, #4695S) and rabbit anti-phospho-ERK1/2 (Thr202/Tyr204) (1:2000, Cell Signalling, #4370S). HRP-conjugated secondary antibodies included goat anti-rabbit (1:2500, Invitrogen, #31460) and goat anti-mouse (1:2500, Invitrogen, #31340) antibodies.

## 3D spheroid viability assay

COV318, A2780, HEY-A8 and OVCAR-8 cells were used for 3D spheroid assays. Cell suspensions were passed through a 0.45 µm filter before seeding 100 µL of cells onto ultra-low attachment (ULA) 96-well black plates with a clear round bottom, at a density of 5000 cells per well. Cells were left for 24-h to allow spheroids to form, and 100 µL of media containing either VC or drug (2× concentration) was added to each well. At endpoint, 150 µL of media was carefully removed from each well and 50 µL of the CellTiter-Glo 3D cell viability assay (Promega, G9682) reagent was added. Cell viability was measured following manufacturer's guidelines. Luminescence was measured on a FLUOstar Omega Microplate reader (BMG LABTECH). Cell-free wells were used as blanks to remove background luminescence. Data were normalised to VC wells.

## Data availability

The authors declare that all data generated or analysed during this study are included in the published article and its supplementary information files. Uncropped blot images can be found in Supplementary Fig. 10. Metabolomics data can be found on figshare: 10.6084/m9.figshare.30218362. Experimental data generated in this study is available in Supplementary Data 1. Flow cytometry gating is show in Supplementary Figs. 11–16.

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

## Acknowledgements

M.G., E.W.T., and H.C.K. acknowledge funding from the EPSRC (EP/S023518/1) and Cancer Research UK Convergence Science Centre (CANTAC721\100021); Y.M. from an Imperial College London and China Scholarship Council joint PhD studentship (202208310101); Yi.X. from an Imperial College London President's PhD Scholarship; L.C.E. from an MRC DTP PhD studentship (MR/W00710X/1); A.N. from an AstraZeneca/NIHR Imperial BRC Imperial College Research Fellowship; A.B. and A.S. from NIHR Imperial BRC; F.M. and J.B. from a STRATiGRAD and MRC DTP PhD studentship (MR/N014103/1); C.Bray from a MRC iCASE Enterprise DTP PhD studentship (MR/R015732/1); Ovarian Cancer Action (A.B., S.S., K.T.,

A.N., C.W. and I.M.). I.M. also acknowledges support from an NIHR Senior Investigator Award. We thank animal technicians from the Imperial College London CBS animal centre for supporting animal studies.

## Author contributions

Conceptualisation: M.G., A.B., I.M. and H.C.K.; Methodology: M.G., S.S., Yi.X., Cr.B., A.B. and H.C.K.; Investigation: M.G., Yi.X., L.A., Y.M. Cr.B., K.T., S.S., Y.X., L.C.E., F.M., J.B., C. W., A.N., A.S., Ch.B. and A.B.; Visualisation: M.G., Yi.X., L.A., Y.M. and A.B.; Funding acquisition: A.B., E.W.T. and H.C.K.; Project administration: A.B., E.W.T. and H.C.K.; Supervision: A.B., E.W.T. and H.C.K.; Writing original draft: M.G. All authors contributed to the review and editing of the paper.

## Competing interests

The authors declare the following competing interests: E.W.T. is a past Director, founder and shareholder in Myricx Bio and consults for and/or receives research funding from Kura Oncology, Pfizer, Samsara Therapeutics, Myricx Pharma, Merck Sharp and Dohme (MSD), Exscientia, Dunad Therapeutics and Daiichi Sankyo. A.B. is currently a paid employee of and has ownership interests at GlaxoSmithKline; however, contributions to this work were made during their employment at Imperial College London. The other authors declare no competing interests.
