## [Transparent Peer Review file · Communications Biology]

RAS/PI3K pathway mutations sensitise epithelial ovarian cancer cells to a PARP/NAMPT inhibitor combination

Corresponding Author: Dr Michael Gruet

Version 0:

Reviewer comments:

Reviewer #1

(Remarks to the Author)

Summary

In this manuscript, the authors aimed to investigate the role of oncogenic mutations in driving the responses to combined inhibition of NAMPT and PARP in epithelial ovarian cancers. The authors took diverse approaches using a panel of ovarian cancer cell lines to identify the specificity and mode of action of the combined inhibition of NAMPT and PARP. They observed specific cytotoxicity in ovarian cancer cells with mutations in the RAS and PI3K pathways. The authors also demonstrated that the combined inhibition of NAMPT and PARP causes greater cell cycle arrest, apoptosis, and growth responses in cell lines with RAS/PI3K mutations.

Review

Strengths: The manuscript's strengths include investigating the potential to expand the use of PARP inhibitors beyond cancers with defects in DNA damage repair pathways and demonstrating the in vivo efficacy of the combined inhibition of NAMPT and PARP using syngeneic mouse models.

Weaknesses: Although the authors demonstrated the cytotoxic effects of NAMPT inhibitor plus PARP inhibitor treatments in cell lines with RAS/PI3K mutations, they require more validation. Below are detailed suggestions to improve the manuscript.

Specific comments:

- (1) The authors demonstrated in Figure 1B that KRAS mutant cell lines are more sensitive to FK866, but they pooled both RAS and PI3K mutated cell lines for their later analyses. Hence, the authors must expand the sensitivity analysis in Figure 1B to PI3K wildtype vs PI3K mutant EOC cells.
- (2) To understand the relationship between RAS/PI3K mutational status and sensitivity to FK866, it is crucial to include the correlation between basal levels of NAD⁺ in EOCs and FK866 responses.
- (3) The authors used COV318 as their RAS/PI3K wildtype control for most of the analyses. However, the growth response data presented in Figure 3A indicates that the basal olaparib sensitivity is lower in COV318 cells when compared to A2780, TOV21G, and OVCAR-8 cells. This raises a concern about whether diminished responses in COV318 are due to the absence of mutations in the RAS/PI3K pathway or additional factors that inhibit basal responses to olaparib in COV318. Additional cell lines that have comparable basal olaparib sensitivity between wildtype and mutant cell lines must be used to measure the changes in cytotoxicity, cell cycle, ROS, apoptosis, and DNA damage with FK866 and Olaparib treatments.
- (4) The authors used a panel of cell lines to demonstrate the correlation between FK866 plus Olaparib treatment sensitivity and RAS/PI3K pathway; however, they did not include any direct evidence that the activity of RAS/PI3K signaling is required for greater sensitivity to FK866 plus Olaparib treatments.
- (5) Although the authors provided a variety of assays to demonstrate the specificity for FK866 plus Olaparib treatments in RAS/PI3K mutated ovarian cancer cells, the underlying mechanisms linking RAS/PI3K signaling and the cellular responses to FK866 plus Olaparib treatment is not clear. The authors must define the mechanisms through which RAS/PI3K signaling could regulate responses to FK866 plus Olaparib treatments.
- (6) Do mutations in RAS/PI3K pathway regulate responses to other PARP inhibitors?
- (7) The results in Figure 3 demonstrate that FK866 sensitizes COV318 cells to olaparib in 2D cultures but not in 3D cultures. The authors must explain how FK866 plus olaparib treatment causes growth arrest in 2D cultures, despite this combination treatment not altering ROS, DNA damage, and apoptosis.
- (8) Dot plots or histograms with gating strategies must be included for all samples in flow cytometry analyses.

Reviewer #2

(Remarks to the Author)

The authors, Grust and Xu, investigated the combination of PARP and NAMPT inhibitors (PARPi/NAMPTi) in epithelial ovarian cancer (EOC), specifically focusing on RAS/PI3K pathway mutant cells. Bioinformatic analysis and screening of EOC cell lines revealed that cells with RAS/PI3K mutations were particularly sensitive to the NAMPT inhibitor FK866. The combination of FK866 and olaparib led to reduced NMN and NAD⁺ levels, increased ROS production, DNA damage, and apoptosis. Additionally, the combination upregulated caspase 3/7 activity, particularly in RAS/PI3K mutant cells. In mouse models, this treatment significantly reduced tumor weight and improved survival, suggesting the potential of the PARPi/NAMPTi combination for treating RAS/PI3K mutant EOC. Below are some suggestions for further strengthening the manuscript and addressing potential limitations:

1. The authors might consider using alternative immortalized ovarian cancer cell lines for their in vitro studies, instead of A2780, which may not be representative of ovarian cancer based on a recent study (PMID: 23839242). This study compared the genomic profiles of cell lines and tumors, concluding that the molecular profiles of commonly used ovarian cancer cell lines differ from those of primary tissues. It would strengthen the manuscript if the authors used other ovarian cancer cell lines, such as OVCAR cells or primary ovarian cell lines, to confirm their in vitro and in vivo results.
2. In Figures 7 and 8, while the combination of PARP and NAMPT inhibitors demonstrated significant efficacy in reducing tumor size and increasing survival in the ID8-Trp53^{-/-}; Pten^{-/-} model, the survival benefit observed (42.5 days compared to 31 days in the vehicle group) is relatively modest. Additionally, the study focuses on a single animal model, and the results may not fully translate to other models, particularly those with different genetic backgrounds or types of ovarian cancer. Including more diverse animal models could help generalize the findings.
3. The study only monitored acute weight changes and survival over a limited timeframe, which does not provide insights into the long-term effects or potential toxicities of the combination therapy. Further studies are needed to explore these aspects and to validate the findings in a broader range of models and clinical settings.
4. In Figures 6 and 3, the authors show that the combination of PARP and NAMPT inhibition only significantly improved PARPi responses in RAS/PI3K-mutant cell lines, not in the RAS/PI3K-wildtype COV318 cell line. This raises concerns about the broader applicability of the treatment across different genetic backgrounds. Further exploration of the mechanisms underlying this selectivity could provide important insights.
5. The authors conducted their study in 2D and 3D spheroid models, but these models may not fully replicate the complexity of in vivo tumor microenvironments. This limits the ability to predict the true effectiveness of the combination therapy in clinical settings. Additional studies using more complex in vivo models could help confirm the relevance of their findings.
6. The combination of PARP and NAMPT inhibitors induced significant apoptosis only in RAS/PI3K-mutant cell lines (A2780 and TOV21G), with little to no effect in the RAS/PI3K-wildtype COV318 cell line. This suggests that the treatment may be selective for certain genetic backgrounds, which could limit its broader applicability. Exploring how different genetic backgrounds influence response to the therapy could help identify which patients might benefit most from this combination treatment.

Reviewer #3

(Remarks to the Author)

In this manuscript, Gruet et al. newly demonstrated that combination treatment with olaparib and FK866 effectively induced cell death in RAS/PI3K-mutated EOC cells. The author's findings are important for the development of future novel therapeutic strategies targeting RAS/PI3K mutant cancers. However, I have several points as indicated below need to be addressed by authors to improve the quality of the articles.

1. In Fig. 3b, COV318 cells as a RAS/PI3K wild-type cell line were used for 3D spheroid formation experiments. However, other wild-type cell lines such as COV644 should also be assessed to clarify the relationship between RAS/PI3K signaling and combined effects of olaparib + FK866.
2. Did you confirm the protein expression levels (including phosphorylated proteins) related to RAS/PI3K signaling in cell lines used for this experiments? The authors should clarify that these pathways were activated in RAS/PI3K mutant cell lines, but not in wild type cell lines.
3. In Figure 4e, the authors showed that the addition of NMN suppressed apoptosis. However, did the authors confirm whether intracellular NMN/NAD levels were upregulated under the condition of NMN supplementation? If you consider that the increase in NMN/NAD levels contributed to the suppression of cell death induced by the combination of olaparib and FK866, this point should be addressed in the manuscript.
4. Did the combined treatment of olaparib and FK866 show synergism? If synergistic effects are observed, this should be mentioned, as it will increase the usefulness for future clinical applications.

5. The authors demonstrated that the combination of olaparib and FK866 induced cell death in RAS/PI3K mutant cancer cells. Several PARP inhibitors, including talazoparib, have been approved as anti-cancer agents. Are these effects specific to olaparib or not?

Version 1:

Reviewer comments:

Reviewer #1

(Remarks to the Author)

The authors have addressed all of my concerns.

Reviewer #3

(Remarks to the Author)

The authors have addressed my concerns and provided satisfactory responses. I have no additional comments.

Combined responses to reviewers

Reviewers' comments:

Reviewer #1 (Remarks to the Author):

Summary

In this manuscript, the authors aimed to investigate the role of oncogenic mutations in driving the responses to combined inhibition of NAMPT and PARP in epithelial ovarian cancers. The authors took diverse approaches using a panel of ovarian cancer cell lines to identify the specificity and mode of action of the combined inhibition of NAMPT and PARP. They observed specific cytotoxicity in ovarian cancer cells with mutations in the RAS and PI3K pathways. The authors also demonstrated that the combined inhibition of NAMPT and PARP causes greater cell cycle arrest, apoptosis, and growth responses in cell lines with RAS/PI3K mutations.

Review

Strengths: The manuscript's strengths include investigating the potential to expand the use of PARP inhibitors beyond cancers with defects in DNA damage repair pathways and demonstrating the in vivo efficacy of the combined inhibition of NAMPT and PARP using syngeneic mouse models.

Weaknesses: Although the authors demonstrated the cytotoxic effects of NAMPT inhibitor plus PARP inhibitor treatments in cell lines with RAS/PI3K mutations, they require more validation. Below are detailed suggestions to improve the manuscript.

Specific comments:

(1) The authors demonstrated in Figure 1B that KRAS mutant cell lines are more sensitive to FK866, but they pooled both RAS and PI3K mutated cell lines for their later analyses. Hence, the authors must expand the sensitivity analysis in Figure 1B to PI3K wildtype vs PI3K mutant EOC cells.

Many thanks for reviewing our manuscript and providing your feedback. We have appropriately amended **figure 1b**, as suggested. This confirms greater FK866 sensitivity is observed in KRAS (*KRAS* only) and PI3K pathway (*PIK3CA* and *PTEN*) mutated cell lines.

(2) To understand the relationship between RAS/PI3K mutational status and sensitivity to FK866, it is crucial to include the correlation between basal levels of NAD⁺ in EOCs and FK866 responses.

We thank the reviewer for their comment, we have updated **figure 2**, basal NAD⁺ levels (and their correlation with FK866 responses) are now shown in **figure 2c**. This demonstrates that RAS/PI3K pathway mutant cell lines have smaller NAD⁺ pools under basal conditions, and these smaller NAD⁺ pools correlate with increased FK866 sensitivity. This is consistent with our other data showing that RAS/PI3K pathway mutant

cell lines reach a critical depletion of NAD⁺ pools more readily with FK866, and hence are more susceptible to FK866 and its combination with Olaparib.

(3) The authors used COV318 as their RAS/PI3K wildtype control for most of the analyses. However, the growth response data presented in Figure 3A indicates that the basal olaparib sensitivity is lower in COV318 cells when compared to A2780, TOV21G, and OVCAR-8 cells. This raises a concern about whether diminished responses in COV318 are due to the absence of mutations in the RAS/PI3K pathway or additional factors that inhibit basal responses to olaparib in COV318. Additional cell lines that have comparable basal olaparib sensitivity between wildtype and mutant cell lines must be used to measure the changes in cytotoxicity, cell cycle, ROS, apoptosis, and DNA damage with FK866 and Olaparib treatments.

We thank the reviewer for their comment, and while we note that COV318 does in fact have comparable sensitivity to Olaparib as at least one RAS/PI3K mutant line (HEY-A8, Fig 3A) that shows substantive response to the combination (Fig. 3b,d & Suppl. Fig. 4a, e), we agree that additional data with wild type cell lines beyond COV318 would be of value to negate specific factors relevant to that line.

To address this we have included several additional key experiments in another RAS/PI3K wildtype control cell line, COV644 (**Fig 4b, Fig 5b**), selected in part on the recommendation of reviewer 3 (see below), but also because these cells lack mutations in other pathways (**Fig 2a**). These new data are consistent with our core argument that the presence/absence of mutation in the RAS/PI3K pathway are a determinant of sensitivity to the combination of NAMPT & PARP inhibition.

(4) The authors used a panel of cell lines to demonstrate the correlation between FK866 plus Olaparib treatment sensitivity and RAS/PI3K pathway; however, they did not include any direct evidence that the activity of RAS/PI3K signaling is required for greater sensitivity to FK866 plus Olaparib treatments.

We thank the reviewer for their comment, and have updated **supplementary figure 1** to include a western blot that highlights basal ERK1/2 phosphorylation and basal AKT phosphorylation (**Supplementary fig. 1b**) in the RAS/PI3K mutant A2780 and TOV21G cell lines, as these cells were used for the majority of our experiments. Furthermore, COV318 and COV644 cell lines were included (RAS/PI3K wild-type). AKT is phosphorylated in A2780 and TOV21G cells. ERK1/2 phosphorylation is detected in all cell lines except for TOV21G cells. Taken together these data highlight that the PI3K/AKT signalling axis is active in A2780 cells (with *BRAF*, *PI3K* and *PTEN* mutations) and TOV21G cells (with *KRAS*, *PI3K* and *PTEN* mutations).

(5) Although the authors provided a variety of assays to demonstrate the specificity for FK866 plus Olaparib treatments in RAS/PI3K mutated ovarian cancer cells, the underlying mechanisms linking RAS/PI3K signaling and the cellular responses to FK866 plus Olaparib treatment is not clear. The authors must define the mechanisms through which RAS/PI3K signaling could regulate responses to FK866 plus Olaparib treatments.

As addressed by additional data in point 2 above, our core mechanistic argument is that RAS/PI3K-mutant cell lines possess smaller NAD⁺ pools (**Fig. 2c and supplementary fig. 2b**) that are more readily depleted by low doses of the NAMPTi FK866 (**Fig. 2d and Supplementary Fig. 2b**); thus dependency on NAMPT activity/sensitivity to FK866 is increased.

As has been well established in the literature (e.g. Bajrami et al., (2012) and Murai et al. (2012)) this will also exacerbate the synthetic lethal interaction observed between a NAMPTi and PARPi combination because: (1) PARP uses NAD⁺ as its substrate, treatment with a PARPi reduces autoPARylation leading to PARP trapping, (2) competition between NAD⁺ and olaparib for binding to the catalytic domain of PARP1/2 occurs but reducing NAD⁺ levels with a NAMPTi increases the binding of olaparib to PARP1/2. (3) This results in persistent DNA lesions and impaired DNA repair, exacerbating the deleterious effects of PARP inhibitors on cells (e.g., increased apoptosis). This has been clarified in the main text [line 348-361].

(6) Do mutations in RAS/PI3K pathway regulate responses to other PARP inhibitors?

We thank the reviewer for their comment, this was outside the scope of what we wanted to achieve during this study as we were primarily interested in the influence of the RAS/PI3K pathway in regulating responses to this specific combination. However, we would anticipate that PARP inhibitors that are similar to olaparib (i.e., strong PARP trappers) such as Niraparib, Talazoparib, Saruparib are likely to exhibit a similar dependency.

It should be noted in genetic backgrounds lacking RAS/PI3K mutations the PARPi/NAMPTi combination has been shown to be generalisable rather than olaparib specific. For example, Heske et al (2017) assessed several PARPis (Niraparib, olaparib and veliparib) and NAMPTis (GNE-618 and GMX-1778). Sauriol et al (2023) also assessed several PARPis (olaparib, niraparib and talazoparib) and NAMPTis (FK866, OT-82 and KPT-9274).

(7) The results in Figure 3 demonstrate that FK866 sensitizes COV318 cells to olaparib in 2D cultures but not in 3D cultures. The authors must explain how FK866 plus olaparib treatment causes growth arrest in 2D cultures, despite this combination treatment not altering ROS, DNA damage, and apoptosis.

We thank the reviewer for their comment, and note that it is common to get differences in sensitivity between 2D and 3D culture. For example, Świerczewska et al., 2023 (<https://doi.org/10.1016/j.biopha.2023.115152>) demonstrate that ovarian cancer cell lines grown as 3D spheroids are more resistant to cytotoxic agents in spheroids than in monolayers. This could be due to difference in gene expression or metabolism in a spheroid vs 2D culture.

The combination of FK866 can sensitise COV318 cells to olaparib (**Fig. 3A and Supplementary Fig.3A**) because it still depletes the NAD⁺ pool (**Fig. 2d, Supplementary Fig. 1c and Supplementary Fig. 2b**), albeit to a lesser extent after 24

hours treatment. This is in line with other studies e.g., Sauriol et al. (2023) (<https://doi.org/10.1038/s41598-023-30081-5>).

However, it should be noted dose response curves (SRB assays) were performed over 6-days. Whereas ROS levels (24hr), MMP loss (48hr) and caspase activation (48hr) were assessed at earlier timepoints. It is likely that changes do occur in ROS/MMP/Caspase in COV318 cells. However, FK866 reduces NAD content in a time- and concentration-dependent manner (Gehrke et al., 2014) (<https://doi.org/10.1158/1078-0432.CCR-14-0624>), and it has been established that NAMPTi is a PARPi sensitiser.

Hence, while we don't dispute that the combination could be effective in models that lack RAS/PI3K mutations, our central message is that lower doses are required to observe synergy in RAS/PI3K-mutant cells, and this is advantageous to prevent NAMPTi-induced toxicity.

(8) Dot plots or histograms with gating strategies must be included for all samples in flow cytometry analyses.

We thank the reviewer for their comment, individual panels hadn't been added in the initial submission to avoid having too many supplementary figure panels. However, dot plots with gating strategies can be found in supplementary data 2.

Reviewer #2 (Remarks to the Author):

The authors, Grust and Xu, investigated the combination of PARP and NAMPT inhibitors (PARPi/NAMPTi) in epithelial ovarian cancer (EOC), specifically focusing on RAS/PI3K pathway mutant cells. Bioinformatic analysis and screening of EOC cell lines revealed that cells with RAS/PI3K mutations were particularly sensitive to the NAMPT inhibitor FK866. The combination of FK866 and olaparib led to reduced NMN and NAD⁺ levels, increased ROS production, DNA damage, and apoptosis. Additionally, the combination upregulated caspase 3/7 activity, particularly in RAS/PI3K mutant cells. In mouse models, this treatment significantly reduced tumor weight and improved survival, suggesting the potential of the PARPi/NAMPTi combination for treating RAS/PI3K mutant EOC. Below are some suggestions for further strengthening the manuscript and addressing potential limitations:

1. The authors might consider using alternative immortalized ovarian cancer cell lines for their in vitro studies, instead of A2780, which may not be representative of ovarian cancer based on a recent study (PMID: 23839242). This study compared the genomic profiles of cell lines and tumors, concluding that the molecular profiles of commonly used ovarian cancer cell lines differ from those of primary tissues. It would strengthen the manuscript if the authors used other ovarian cancer cell lines, such as OVCAR cells or primary ovarian cell lines, to confirm their in vitro and in vivo results.

We thank the reviewer for their feedback. It is important to note that the A2780 cell line is a model of endometrioid ovarian cancer (Tudrej et al., 2018) but was previously misclassified as high grade serous ovarian cancer. In the present study we were

assessing the efficacy of the combination in all epithelial ovarian cancer subtypes, as RAS/PI3K mutations occur in each histological subtype.

The concerns regarding A2780 cells notwithstanding, we use several other lines recognised in the cited study by Domcke et al., (2013) as *bone fide* ovarian cancer models. As noted when addressing point 3, reviewer 1 above, we have also included substantive new data in COV644 cells, an additional ovarian cancer model.

Unfortunately, we did not have access to suitable primary ovarian models.

2. In Figures 7 and 8, while the combination of PARP and NAMPT inhibitors demonstrated significant efficacy in reducing tumor size and increasing survival in the ID8-Trp53^{-/-}; Pten^{-/-} model, the survival benefit observed (42.5 days compared to 31 days in the vehicle group) is relatively modest. Additionally, the study focuses on a single animal model, and the results may not fully translate to other models, particularly those with different genetic backgrounds or types of ovarian cancer. Including more diverse animal models could help generalize the findings.

We thank the reviewer for their comment, although we think it is important to highlight that the ID8-Trp53^{-/-}; Pten^{-/-} model is a highly aggressive model (Walton et al., 2017). Despite this the combination increased median survival by over 30%, more than double the benefit observed with a PARPi alone with no additional toxicity, which we regard as substantive.

While we do agree it is valuable to assess whether the combination will translate to other *in vivo* models this was unfortunately out of the scope of what we could achieve during this project.

3. The study only monitored acute weight changes and survival over a limited timeframe, which does not provide insights into the long-term effects or potential toxicities of the combination therapy. Further studies are needed to explore these aspects and to validate the findings in a broader range of models and clinical settings.

We thank the reviewer for their comment, however we also note that there were no changes in behaviour or signs of pain were displayed by mice throughout each study. Furthermore, after mice were culled upon reaching a humane endpoint no signs of toxicity were observed (e.g., appearance/weight of organs). Thus while we do agree it is essential to further assess the long-term effects or potential toxicities of the PARPi/NAMPTi combination additional studies were beyond the scope of this project.

4. In Figures 6 and 3, the authors show that the combination of PARP and NAMPT inhibition only significantly improved PARPi responses in RAS/PI3K-mutant cell lines, not in the RAS/PI3K-wildtype COV318 cell line. This raises concerns about the broader applicability of the treatment across different genetic backgrounds. Further exploration of the mechanisms underlying this selectivity could provide important insights.

We thank the reviewer for their comment, and note the mechanistic rationale provided in response to point 5, reviewer 1. While we have shown the combination to be more effective in RAS/PI3K-mutant cell lines, we note that ~45% of high-grade serous carcinoma of the ovary (the most common ovarian cancer subtype) cases have dysregulated RAS/PI3K-pathway signalling through mutations and copy number alterations in these pathways. This could substantially expand the range of genetic backgrounds for which the combination is relevant. This point is noted in the main text [lines 359-361].

5. The authors conducted their study in 2D and 3D spheroid models, but these models may not fully replicate the complexity of *in vivo* tumor microenvironments. This limits the ability to predict the true effectiveness of the combination therapy in clinical settings. Additional studies using more complex *in vivo* models could help confirm the relevance of their findings.

We thank the reviewer for their comment, and while we note that the model used is immunocompetent and can mimic the immune tumour microenvironment, we agree that additional *in vivo* models would be valuable. Unfortunately, it was outside the scope of this study to perform additional experiments in more *in vivo* models.

6. The combination of PARP and NAMPT inhibitors induced significant apoptosis only in RAS/PI3K-mutant cell lines (A2780 and TOV21G), with little to no effect in the RAS/PI3K-wildtype COV318 cell line. This suggests that the treatment may be selective for certain genetic backgrounds, which could limit its broader applicability. Exploring how different genetic backgrounds influence response to the therapy could help identify which patients might benefit most from this combination treatment.

We thank the reviewer for their comment, we refer to our response to point 4 above.

Reviewer #3 (Remarks to the Author):

In this manuscript, Gruet et al. newly demonstrated that combination treatment with olaparib and FK866 effectively induced cell death in RAS/PI3K-mutated EOC cells. The author's findings are important for the development of future novel therapeutic strategies targeting RAS/PI3K mutant cancers. However, I have several points as indicated below need to be addressed by authors to improve the quality of the articles.

1. In Fig. 3b, COV318 cells as a RAS/PI3K wild-type cell line were used for 3D spheroid formation experiments. However, other wild-type cell lines such as COV644 should also be assessed to clarify the relationship between RAS/PI3K signalling and combined effects of olaparib + FK866.

We thank the reviewer for their feedback. Unfortunately, in our hands COV644 cells did not form 3D spheroids. However, we have included substantive additional data using COV644 cell line in other key experiments (**Fig. 4b and Fig. 5b**), all of which are consistent with our original observations in the COV318 cell line.

2. Did you confirm the protein expression levels (including phosphorylated proteins) related to RAS/PI3K signaling in cell lines used for this experiments? The authors should clarify that these pathways were activated in RAS/PI3K mutant cell lines, but not in wild type cell lines.

We thank the reviewer for their comment, we have updated **supplementary figure 1** to include a western blot that probes for ERK1/2 and AKT phosphorylation under basal conditions (**Supplementary fig. 1b**). The RAS/PI3K-mutant A2780 and TOV21G cells were included as these cells were used in the majority of our experiments. Several RAS/PI3K-wildtype cell lines were included (COV318 and COV644). We observed that AKT is phosphorylated in A2780 and TOV21G (highest levels) cells. Whereas ERK1/2 phosphorylation was detected in all cell lines except for TOV21G cells. Taken together, these data highlight that the PI3K/AKT signalling axis is active in A2780 cells (with *BRAF*, *PI3K* and *PTEN* mutations) and TOV21G cells (with *KRAS*, *PI3K* and *PTEN* mutations).

3. In Figure 4e, the authors showed that the addition of NMN suppressed apoptosis. However, did the authors confirm whether intracellular NMN/NAD levels were upregulated under the condition of NMN supplementation? If you consider that the increase in NMN/NAD levels contributed to the suppression of cell death induced by the combination of olaparib and FK866, this point should be addressed in the manuscript.

We thank the reviewer for their comment, we did not directly assess whether NMN/NAD⁺ levels were upregulated under the condition of NMN supplementation in these experiments. However, the observed rescue with NMN supplementation in caspase (**Fig. 4e-f**) and dose response experiments (**Fig. 7a**) would suggest we are upregulating intracellular NMN/NAD⁺ levels.

We note that in another study Draganov et al., (2024) (<https://doi.org/10.1039/D4CB00043A>) we did confirm that supplementation with the selected dose of NMN upregulates the intracellular concentration of NMN/NAD⁺ in a range of cell lines (e.g., HEK293T).

4. Did the combined treatment of olaparib and FK866 show synergism? If synergistic effects are observed, this should be mentioned, as it will increase the usefulness for future clinical applications.

We thank the reviewer for their comment. Indeed, synergy is observed following treatment with the combination, with stronger synergy observed in RAS/PI3K mutant cell lines. The text has been updated to reflect this [lines 140-143] and synergy plots have been added (**Supplementary fig. 4**).

5. The authors demonstrated that the combination of olaparib and FK866 induced cell death in RAS/PI3K mutant cancer cells. Several PARP inhibitors, including talazoparib, have been approved as anti-cancer agents. Are these effects specific to olaparib or not?

We thank the reviewer for their comment. Although, in genetic backgrounds lacking RAS/PI3K mutations the PARPi/NAMPTi combination has been shown to be

generalisable rather than olaparib specific. For example, Heske et al (2017) assessed several PARPis (Niraparib, olaparib and veliparib) and NAMPTis (GNE-618 and GMX-1778). Sauriol et al (2023) also assessed several PARPis (olaparib, niraparib and talazoparib) and NAMPTis (FK866, OT-82 and KPT-9274).